# Muscarinic receptors mediate motivation via preparatory neural activity in humans

John P Grogan[1,2]*, Matthias Raemaekers[3], Maaike MH van Swieten[1], Alexander L Green[1], Martin J Gillies[4], Sanjay G Manohar[1]

[1]Nuffield Department of Clinical Neuroscience, University of Oxford, Oxford, United Kingdom; [2]Trinity College Institute of Neuroscience and Department of Psychology, Trinity College Dublin, Dublin, Ireland; [3]Department of Experimental, Clinical and Health Psychology, Ghent University, Ghent, Belgium; [4]Nuffield Department of Surgical Sciences, University of Oxford, Oxford, United Kingdom

## eLife Assessment

The authors have reported an **important** study in which they use a double-blind design to explore pharmacological manipulations in the context of a behavioral task. While the sample size is small, the use of varied methodology, including electrophysiology, behavior, and pharmacology, makes this manuscript particularly notable. Overall, the findings are **solid** and motivate future explanations into the relationships between acetylcholine and motivation.

*For correspondence:
john.grogan@tcd.ie

**Competing interest:** The authors declare that no competing interests exist.

**Abstract** Motivation depends on dopamine, but might be modulated by acetylcholine which influences dopamine release in the striatum, and amplifies motivation in animal studies. A corresponding effect in humans would be important clinically, since anticholinergic drugs are frequently used in Parkinson's disease, a condition that can also disrupt motivation. Reward and dopamine make us more ready to respond, as indexed by reaction times (RT), and move faster, sometimes termed vigour. These effects may be controlled by preparatory processes that can be tracked using electroencephalography (EEG). We measured vigour in a placebo-controlled, double-blinded study of trihexyphenidyl (THP), a muscarinic antagonist, with an incentivised eye movement task and EEG. Participants responded faster and with greater vigour when incentives were high, but THP blunted these motivational effects, suggesting that muscarinic receptors facilitate invigoration by reward. Preparatory EEG build-up (contingent negative variation [CNV]) was strengthened by high incentives and by muscarinic blockade, although THP reduced the incentive effect. The amplitude of preparatory activity predicted both vigour and RT, although over distinct scalp regions; frontal activity predicted vigour, whereas a larger, earlier, central component predicted RT. The incentivisation of RT was partly mediated by the CNV, though vigour was not. Moreover, the CNV mediated the drug's effect on dampening incentives, suggesting that muscarinic receptors underlie the motivational influence on this preparatory activity. Taken together, these findings show that a muscarinic blocker impairs motivated action in healthy people, and that medial frontal preparatory neural activity mediates this for RT.

## Introduction

Motivation is our ability to exert effort to obtain reward, and can be dramatically impacted in psychiatric and neurological disorders. One simple index of motivation is response vigour: an increase in movement speed with reward (*Dudman and Krakauer, 2016*; *Shadmehr et al., 2010*). Motivation and the lack of motivation (apathy) have been linked to dopamine (*McGuigan et al., 2019*; *Walton*

*and Bouret, 2019*), and accordingly, vigour is reduced in Parkinson's disease (*Beierholm et al., 2013*; *da Silva et al., 2018*; *Mazzoni et al., 2007*; *Zénon et al., 2016*).

Striatal dopamine levels ramp up before potentially rewarding actions. Animal work demonstrates that acetylcholine controls these dopaminergic effects, via both firing rates (*Forster and Blaha, 2000*; *Mark et al., 2011*) and local release in the striatum (*Cachope and Cheer, 2014*; *Shen et al., 2007*). These provide two mechanisms for acetylcholine to affect motivation (*Collins et al., 2019*; *Hoebel et al., 2007*), with muscarinic receptors in nucleus accumbens playing an important role in facilitating reward-related vigour (*Collins et al., 2016*), possibly by potentiating dopamine release and affecting the frequency sensitivity of striatal dopaminergic terminals (*Collins et al., 2016*; *Threlfell et al., 2010*). However, the ultimate effect of systemic muscarinic drugs on motivation is complex. Muscarinic antagonism has impaired motivation in some animal studies (*Collins et al., 2016*; *Ostlund et al., 2014*; *Pratt and Kelley, 2004*), while improving it in others (*Hailwood et al., 2019*; *Nunes et al., 2013*), and even less is known of how they affect humans. Pro-cholinergic drugs may ameliorate clinical apathy, a disabling symptom seen in around 50% of patients with Parkinson's disease (*Devos et al., 2014*; *Fahed and Steffens, 2021*; *Pagonabarraga et al., 2015*), yet cholinergic *blockers* are commonly used to treat the motor symptoms. This frequently puts clinicians in a dilemma when treating patients with both tremor, which responds well to anticholinergic drugs such as trihexyphenidyl (THP), but also apathy and cognitive impairment, which both respond to pro-cholinergic drugs (*Bohnen et al., 2022*). Distinguishing the mechanisms by which acetylcholine receptors contribute to movement and motivation will be critical for selecting appropriate treatments in these patients. A first step would be to test whether the effects of muscarinic blockade on motivation, observed in animals, generalise to humans.

Motivation influences action selection and movement invigoration based on the motivational state that is set up before the action. One potential mechanism of this may be preparatory activation of frontal premotor areas, which may be indexed on electroencephalography (EEG) by the contingent negative variation (CNV; *Walter et al., 1964*), a slow negative potential that appears between a warning stimulus and a prompt to act (*Brunia et al., 2012*). Early models of the CNV proposed that it reflected cholinergic activity, modulated by dopamine, noradrenaline, and GABA (*Timsit-Berthier, 1991*). Supporting this, muscarinic antagonists were found to disrupt the CNV in rodents (*Ebenezer, 1986*; *Papart et al., 1997*; *Pirch et al., 1986*; *Papart et al., 1997*). This anticipatory signal reflects preparatory activation of supplementary motor area and anterior cingulate, that can be amplified by reward signals from ventral striatum (*Nagai et al., 2004*; *Plichta et al., 2013*). As a marker of motivation, the CNV is of great clinical interest, being decreased by depression (*Ansseau et al., 1985*) and Parkinson's disease (*Ikeda et al., 1997*), and conversely, increased by dopaminergic medications (*Linssen et al., 2011*). Accordingly, the CNV is strengthened by monetary incentives dependent on performance (*Berchio et al., 2019*; *Novak et al., 2016*; *Novak and Foti, 2015*) and reward contingency (*Frömer et al., 2021*). Changes in the CNV could therefore provide a mechanistic handle on the effect of drugs on motivation.

Here, we ask whether blocking muscarinic receptors would reduce motivation and increase distractibility in humans, and further, whether this is mediated by preparatory activity in medial frontal areas. To test this, we measured the CNV in healthy adults after administration of an anti-muscarinic M1 receptor (M1r) acetylcholine antagonist (THP) or placebo (double-blinded), while they performed an incentivised eye movement task. We used a task that independently measured action selection and energisation, which may involve different neural mechanisms.

## Results

After fixating on one of three circles, participants heard an incentive cue stating the maximum reward available for the trial (50p or 0p), and 1500 ms after a preparation cue they made a saccade towards the circle that dimmed, and received a proportion of the maximum reward depending on their speed. On half of the trials, the non-target circle lit up, presenting a high-salience distractor. Participants completed this task once after receiving THP (M1r antimuscarinic acetylcholine antagonist) and once after placebo (double-blinded), and we measured saccade initiation times, velocity, and the pull of the distractor on the saccade trajectory, along with EEG.

## Acetylcholine modulates invigoration by incentives

We measured vigour as the residual peak velocity of saccades within each drug session (see *Figure 1c* and Methods/Eye-tracking), which is each trial's deviation of velocity from the main sequence. This removes any overall effects of the drug on saccade velocity, while still allowing incentives and distractors to have different effects within each drug condition. We used single-trial mixed-effects linear regression (20 participants, 18,585 trials in total) to assess the effects of incentive, distractors, and THP, along with all the interactions of these (and a random intercept per participant), on residual velocity and saccadic RT. As predicted, residual peak velocity was increased by incentives (*Figure 1d*; β=0.1266, p<0.0001), while distractors slightly slowed residual velocity (β=–0.0158, p=0.0294; see *Table 1* for full behavioural statistics). THP decreased the effect of incentives on velocity (incentive * THP: β=–0.0216, p=0.0030), indicating that muscarinic blockade diminished motivation by incentives. *Figure 1d* shows that this effect was similar in distractor absent/present trials, although slightly stronger when the distractor was absent; the three-way (distractor*incentive*THP) interaction was not significant (p>0.05), suggesting that the distractor-present trials had the same effect but weaker (*Figure 1d*).

Saccadic RT (time to initiation of saccade) was slower when participants were given THP (β=0.0244, p=<0.0001), faster with incentives (*Figure 1e*; β=–0.0767, p<0.0001), and slowed by distractors (β=0.0358, p<0.0001). Again, THP reduced the effects of incentives (incentive*THP: β=0.0218, p=0.0002). *Figure 1e* shows that this effect was similar in distractor absent/present trials, although slightly stronger when the distractor was present; as the three-way (distractor*incentive*THP) interaction was not significant and the direction of effects was the same in the two, it suggests the effect was similar in both conditions. Additionally, the THP*incentive interactions were correlated between saccadic RT and residual velocity at the participant level (*Figure 1—figure supplement 1*).

## Cholinergic blockade increases distractibility

We measured distractibility as the angular deviation of the eye position away from the target's orientation towards the distractor's location, at the start of the saccade (*Figure 2a*), which indicates the pull of the distractor. The distribution of distractor pull is bimodal, with saccades directed towards either the target or distractor locations (*Figure 2c*). When the distractor did not light up, the mean distractor angle was negative (i.e. away from the distractor angle), indicating repulsion away from the angle of the distractor, whereas when the distractor lit up and was salient the mean angle was biased towards the distractor (*Figure 2b*; β=0.2446, p<0.0001; see *Table 1*). Single-trial linear mixed-effects regression found that THP increased the pull (main effect of drug, β=0.0283, p<0.0001), but only when the distractor was salient (THP*distractor, β=0.0226, p=0.0012, pairwise drug effect: distractor absent: p>0.3; present: β=0.0511, p<0.0001). Unlike in previous work, we found no effect of incentives on distraction (β=0.0023, p=0.7444), although speed-accuracy trade-off curves (*Figure 2—figure supplement 1*) showed that incentives sped up responses in such a way that distraction was reduced for a given RT.

The drug-related increase in distraction could be due to either greater pull or reduced repulsion. To distinguish these possibilities, we plotted the distribution of distractor pull across trials where the distractor was present (*Figure 2c*). THP reduced the probability of repulsion (*Figure 2d*) around –30°, indicating that THP reduced the repulsion away from the distractor's location. This suggests weaker attentional suppression of the distractor. Incentives had little effect on the distribution (yellow line) in keeping with the averages in *Figure 2b*.

Therefore, acetylcholine antagonism reduced the invigoration of saccades by incentives, and decreased the repulsion of salient distractors. We next asked whether these effects were coupled with changes in preparatory neural activity.

## Preparatory neural activity is modulated by incentive and acetylcholine

We examined EEG activity in the delay period between the preparation cue and the target (and distractor) onset, first using three time-windows of interest, then a cluster-based permutation approach. There was an early fronto-central positive event-related potential (ERP) with a peak around 220 ms, consistent with the P3a (*Figure 3a*), which was then followed by a growing negative potential centrally, consistent with the CNV. From the grand-average ERP over all conditions, we chose 200–280

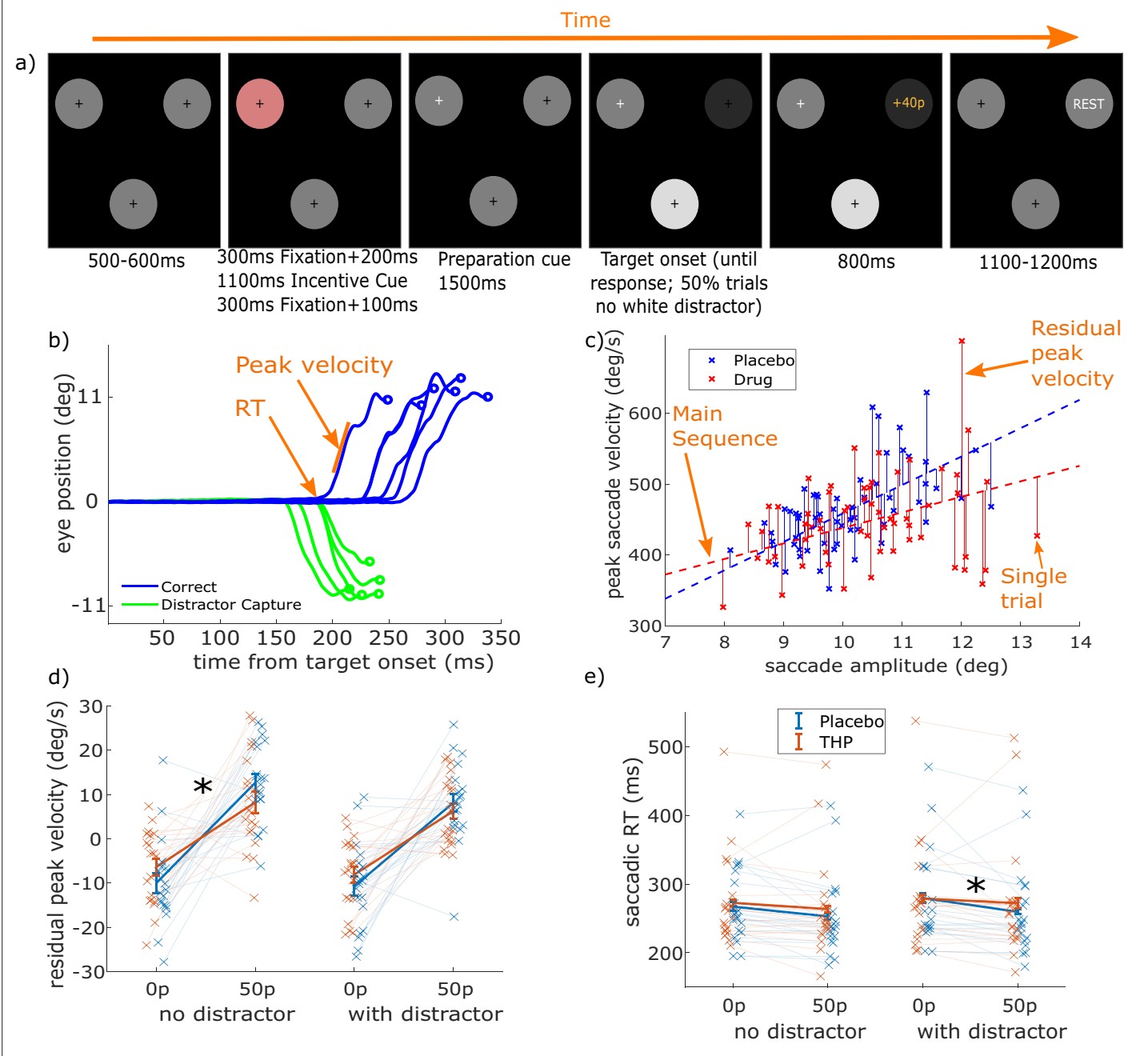

**Figure 1.** Trihexyphenidyl (THP) modulates saccadic measures. (**a**) Trial structure for a high incentive trial with a salient distractor. After fixation on the pink starting circle, the incentive cue plays, and after a short fixation wait the preparation cue is given which is the fixation cross turning white. 1500 ms later, one circle dims, which is the target, and on 50% of trials the other circle brightens (salient distractor). Feedback is given when participants saccade to the target, based on their speed. Timings are given below each screen. (**b**) Eye position as a function of time for a selection of saccades. Saccade reaction time (RT) is the time at which the saccade begins, peak velocity is the maximal speed during the movement (steepest slope here), and amplitude is the distance to the saccade endpoint. (**c**) Plotting peak velocity against amplitude (sample data) shows the main sequence effect (dashed lines) where larger saccades have higher velocity. We regressed velocity on amplitude separately within each drug session (to control for potential drug effects on amplitude, velocity, or the main sequence), giving residual peak velocity as our measure of vigour (solid vertical lines). (**d**) Mean peak residual velocity for each condition (20 participants, 18585 trials). Incentives increased velocity (single-trial linear mixed-effects regression; β=0.1266, p<0.0001; see *Table 1* for full statistics), distractors decreased it (β=−0.0158, p=0.0294), and THP reduced the invigoration by incentives (β=−0.0216, p=0.0030). This interaction was significant only for the no-distractor trials. Crosses show individual participant means for each condition, and error bars show within-subject SEM. (**e**) Mean saccadic RT for each condition (log RT was analysed, raw values plotted here). High incentives decreased RT (β=−0.0767,

*Figure 1 continued on next page*

*Figure 1 continued*

p<0.0001), distractors slowed RT (β=0.0358, p<0.0001), and THP reduced the effect of incentive on RT (β=0.0218, p=0.0002) – which was driven by trials with distractors present. Crosses show individual participant means for each condition.

The online version of this article includes the following source data, source code, and figure supplement(s) for figure 1:

**Source code 1.** Matlab code to produce *Figure 1c–e* (see GitHub repo for additional required functions).

**Source data 1.** mat file with data to produce *Figure 1c–e*.

**Figure supplement 1.** Incentive effects are uncorrelated between saccade reaction times (RT) and velocity, but the THP*incentive interactions are correlated.

**Figure supplement 1—source code 1.** Matlab code to produce *Figure 1—figure supplement 1*.

ms at Cz for the early ERP, and 1200–1500 ms at Cz for the CNV. Note that both these periods began >1.5 s after the incentive was presented.

We used single-trial linear mixed-effects regression to see the effects of incentive and THP on each ERP (20 participants, 16,627 trials; distractor was included too, along with all interactions, and a random intercept by participant). Prior to the preparation cue (900–1100 ms after incentive cue, baselining at the incentive cue; green shaded area in *Figure 3a*), THP strengthened negativity (*Figure 3b*, β=–0.0597, p<0.0001; see *Table 2* for full ERP statistics), but incentives had no effect or interaction (p>0.05). After the preparation cue, the P3a (*Figure 3c*) was significantly smaller for high incentive trials (β=–0.0187, p=0.0142) with no other significant effects (p>0.1). The subsequent

**Table 1.** Linear mixed-effects single-trial regression outputs for behavioural variables.

Each model included a random effect of participant (20 participants, 18585 trials), along with all lower-order interactions and main effects: 'behaviour ~ 1 + incentive * distractor * THP + (1 | participant)'. RT was log-transformed for this analysis. Significant effects are shown in bold italics.

| Measure | Term | β | CI | SE | t | p |
|---|---|---|---|---|---|---|
| | *Incentive* | *0.1266* | *0.1123, 0.1408* | *0.0073* | *17.4001* | *<0.0001* |
| | *Distractor* | *–0.0158* | *–0.0301,–0.0016* | *0.0073* | *–2.1786* | *0.0294* |
| | THP | –0.0001 | –0.0144, 0.0141 | 0.0073 | –0.0153 | 0.9878 |
| | Incentive * distractor | –0.0067 | 0.0201, 0.0076 | 0.0073 | –0.9143 | 0.3605 |
| | *Incentive * THP* | *–0.0216* | *–0.0358,–0.0073* | *0.0073* | *–2.9678* | *0.0030* |
| | Distractor * THP | 0.0023 | –0.0120, 0.0165 | 0.0073 | 0.3152 | 0.7526 |
| Residual velocity (df = 1, 18,577) | Incentive * distractor * THP | 0.0052 | –0.0091, 0.0195 | 0.0073 | 0.7158 | 0.4741 |
| | *Incentive* | *–0.0767* | *–0.0884,–0.0651* | *0.0059* | *–12.9162* | *<0.0001* |
| | *Distractor* | *0.0348* | *0.0231, 0.0464* | *0.0059* | *5.8549* | *<0.0001* |
| | *THP* | *0.0244* | *0.0127, 0.0360* | *0.0059* | *4.1010* | *<0.0001* |
| | Incentive * distractor | –0.0035 | –0.0151, 0.0082 | 0.0059 | –0.5826 | 0.5601 |
| | *Incentive * THP* | *0.0218* | *0.0102, 0.0335* | *0.0059* | *3.6723* | *0.0002* |
| | *Distractor * THP* | *–0.0117* | *–0.0233,–0.0001* | *0.0059* | *–1.9689* | *0.0490* |
| Saccade RT (df = 1, 18,577) | Incentive * distractor * THP | 0.0076 | –0.0041, 0.0192 | 0.0059 | 1.2714 | 0.2036 |
| | Incentive | 0.0023 | –0.0114, 0.0160 | 0.0070 | 0.3261 | 0.7444 |
| | *Distractor* | *0.2446* | *0.2309, 0.2583* | *0.0070* | *35.0416* | *<0.0001* |
| | *THP* | *0.0283* | *0.0146, 0.0420* | *0.0070* | *4.0570* | *<0.0001* |
| | Incentive * distractor | 0.0028 | –0.0109, 0.0165 | 0.0070 | 0.3982 | 0.6905 |
| | Incentive * THP | 0.0030 | –0.0107, 0.0167 | 0.0070 | 0.4340 | 0.6643 |
| | *Distractor * THP* | *0.0226* | *0.0089, 0.0363* | *0.0070* | *3.2348* | *0.0012* |
| Distractor pull (df = 1, 18,577) | Incentive * distractor * THP | –0.0039 | –0.0177, 0.0098 | 0.0070 | –0.5631 | 0.5734 |

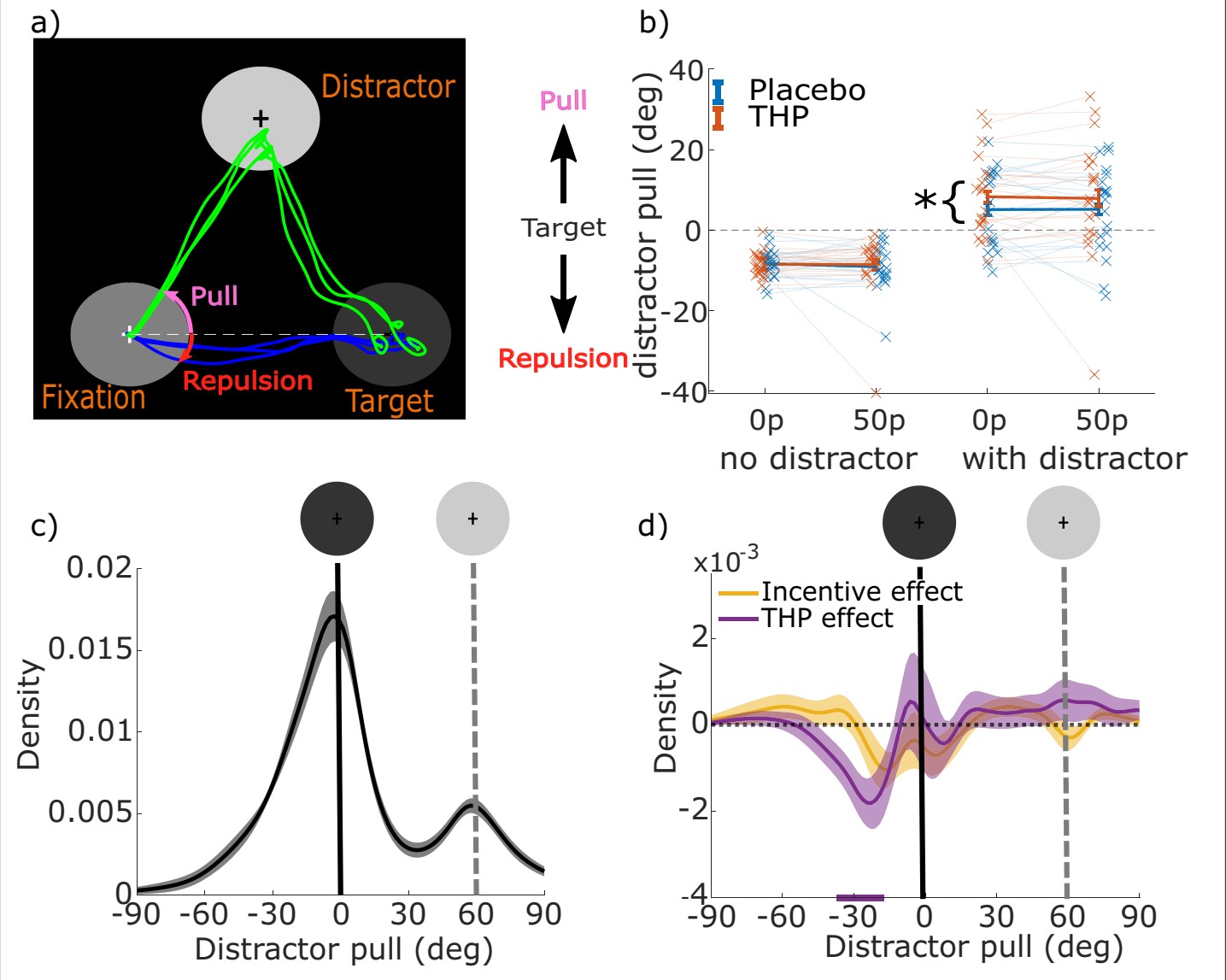

**Figure 2.** Muscarinic blockade increases the pull from salient distractors. (**a**) Sample saccades showing fixation at the bottom left circle, the target on the right, and the distractor on the top. Distractor pull is the angle of the eye when it leaves the fixation circle, relative to a straight line from the fixation to target circle (positive values reflect angles towards the distractor, zero is flat, negative reflects repulsion). (**b**) Mean distractor pulls for low and high incentives when the salient distractor is and is not present (crosses show individual participant means per condition, error-bars are within-subject SEM; 20 participants, 18585 trials). Distractor pull was negative (i.e. below the horizontal line in panel a) reflecting repulsion from the distractor when it did not light up. However, when the distractor did light up, distractor pull was positive (single-trial linear mixed-effects regression; β=0.2446, p<0.0001), reflecting a bias towards it, and this bias was greater on trihexyphenidyl (THP) than placebo (distractor*THP interaction: β=0.0226, p=0.0012; full statistics are given in *Table 1*). (**c**) Mean kernel-smoothed density of distractor pulls for all trials with a distractor (averaged across all other conditions) with shading showing the within-subject standard errors. There is a smaller peak centred on the distractor's orientation (grey dashed line and circle). Negative distractor pulls show the repulsive bias away from the distractor location. (**d**) Mean kernel-smoothed densities showing the effects of incentive (i.e. 50p – 0p) and THP (i.e. THP – incentive) for all 'with distractor' trials. Cluster-based permutation testing showed that THP reduced the number of trials biased around –30° (p<0.05), indicating reduced repulsive bias when muscarinic receptors are antagonised.

The online version of this article includes the following source data, source code, and figure supplement(s) for figure 2:

**Source code 1.** Matlab code to produce *Figure 2b–d* (see GitHub repo for additional required functions).

**Source data 1.** mat file with data for *Figure 2b–d*.

**Figure supplement 1.** Distractor pull as a function of reaction time (RT).

**Figure supplement 1—source code 1.** Matlab file to produce *Figure 2—figure supplement 1*.

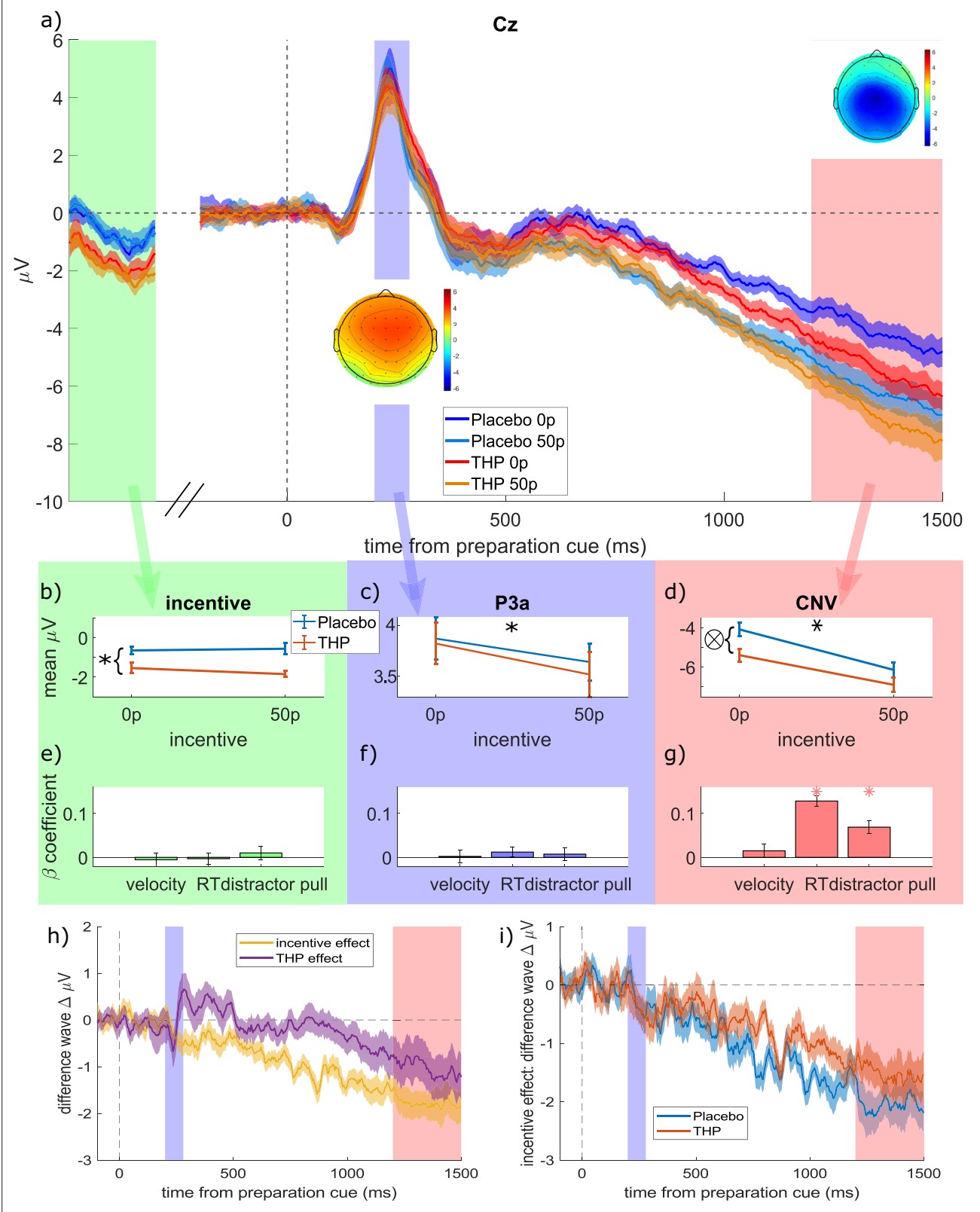

**Figure 3.** Mean event-related potentials (ERPs) to the preparation cue. (**a**) Grand-average ERPs in electrode Cz split for the four conditions (low and high incentive, placebo and trihexyphenidyl [THP]; 20 participants, within-subject SEM error-bars). The three time-windows are highlighted in different colours, and correspond to the columns of panels below, with topographies of the mean amplitude within each window superimposed. The 'incentive' window is a non-contiguous window of 900–1100 ms after the incentive cue, just before the preparation cue appears, which contains the late

*Figure 3 continued*

negative potential after the incentive cue. (**b–d**) The mean voltages within each time-window for the different incentive and drug conditions (individual participants' data are shown in *Figure 3—figure supplement 1*, and full statistics are given in *Table 2*). (**b**) Late ERP to the incentive cue (900:1100 ms at Cz) was more negative when on THP than placebo (single-trial linear mixed-effects regression with 16627 trials; β=–0.0597, p<0.0001), but it was not affected by incentive (p>0.05). (**c**) Mean P3a (200:280 ms after the preparation cue) is decreased by high incentives (β=–0.0187, p=0.0142) but unaffected by THP (p>0.1, note the different y-axis scale to b and d). (**d**) The contingent negative variation (CNV) (1200:1500 ms after the preparation cue) is strengthened (more negative) by incentives (β=–0.0928, p<0.0001) and THP (β=–0.0502, p<0.0001), with a weak interaction (β=0.0172, p=0.0213) as THP slightly reduces the incentive effect (flatter slope for the orange line; and THP lines are closer than placebo lines in panel a). (**e–g**) The beta-coefficients from regressing each component against each behavioural variable, with stars representing significant associations (p<0.0056; Bonferroni-corrected for nine comparisons, error bars are 95% CI). (**h**) Difference waves showing the effects of incentive (50p – 0p, averaged over other factors) and THP (THP – placebo, averaged over other factors). Incentive starts decreasing (i.e. strengthening) the CNV early (during the P3a window), while the THP effect starts around 900 ms after the preparation cue. (**i**) Difference waves showing incentive effects within each drug condition separately. Incentives strengthen the CNV for both conditions, with the effect growing more slowly for the THP condition, reflecting the THP*incentive interaction reported in the main text.

The online version of this article includes the following source data, source code, and figure supplement(s) for figure 3:

**Source code 1.** Matlab code to produce figure panels (see GitHub repo for additional required functions).

**Source data 1.** mat file with data to produce figure panels, including *Figure 3—figure supplement 1*.

**Source data 2.** Linear mixed-effects single-trial regression outputs for the effect of pre-preparation cue activity, P3a, and contingent negative variation (CNV) on each behavioural measure.

**Figure supplement 1.** Mean individual participants' event-related potential (ERP) component voltages for each time-window.

**Figure supplement 2.** Testing for effects of incentives, trihexyphenidyl (THP), or THP*incentives across all electrodes and time-points.

**Figure supplement 2—source data 1.** mat file with data to produce *Figure 3—figure supplement 2*.

**Figure supplement 2—source code 1.** Matlab file to produce *Figure 3—figure supplement 2*.

CNV was strengthened (i.e. more negative; *Figure 3d*) by incentive (β=–0.0928, p<0.0001) and THP (β=–0.0502, p<0.0001), with an interaction whereby *THP decreased the incentive effect* (β=0.0172, p=0.0213). *Figure 3h* shows the effects of incentive and THP on the CNV separately, using difference waves, and *Figure 3i* shows the incentive effect grows more slowly in the THP condition than the placebo condition.

This suggests that while incentives strengthened the incentive-cue response and the CNV and weakened the P3a, muscarinic antagonism strengthened the CNV, and the incentive*THP interaction was only seen on the CNV in the same direction as that seen with the vigour and RT – THP reduced the incentive effect. Thus, although the drug and reward both increased the CNV build-up, the drug reduced the reward effect.

## Neural preparation predicts RT and distraction

We regressed the mean amplitude of the pre-preparation activity, P3a and CNV against velocity, RT, and distractor pull (including incentive, distractor, and drug conditions as covariates), and Bonferroni-corrected the p-values for nine multiple comparisons. Pre-preparation voltage and P3a did not predict any behavioural variable (*Figure 3e–f*), while CNV predicted RT and distractor pull (*Figure 3g*, p<0.0001; see *Figure 3—source data 2* for statistics) but not vigour.

As vigour was not associated with our a priori ERPs, we ran a window-free analysis to find spatial or temporal regions of the EEG activity that was associated with vigour, as well as RT and distractor pull. This is an exploratory analysis, and given the small sample size should be interpreted with caution. We regressed each electrode and time-point against the three behavioural variables separately, while controlling for effects of incentive, distractor, THP, the interactions of those factors, and a random effect of participant. This analysis therefore asks whether trial-to-trial neural variability predicts behavioural variability. To assess significance, we used cluster-based permutation tests (DMGroppe Mass Univariate toolbox; *Groppe et al., 2011*), shuffling the trials within each condition and person, and repeating it 2500 times, to build a null distribution of 'cluster mass' from the t-statistics (*Bullmore et al., 1999*; *Maris and Oostenveld, 2007*) which was used to calculate two-tailed p-values with a family-wise error rate (FWER) of 0.05 (see Methods/Analysis for details). Velocity, RT, and distraction were predicted by preparatory EEG voltages before the onset of the target, each with distinct patterns (*Figure 4*). Residual velocity was significantly predicted by frontal electrodes from about 280 ms after the preparation cue, which was strongest on electrode AF8. This did not encompass electrode Cz. RT

**Table 2.** Linear mixed-effects single-trial regression outputs for P3a and contingent negative variation (CNV).
Significant effects are shown in bold italics. Each model also included a random effect of participant, along with all lower-order interactions and main effects: 'ERP ~ 1 + incentive * distractor * THP + (1 | participant)'. There were 20 participants and 16627 trials in total.

| Measure | Term | β | CI | SE | t | p |
|---|---|---|---|---|---|---|
| P3a (df = 1, 16,619) | *Incentive* | *–0.0186* | *–0.0335,–0.0037* | *0.0076* | *–2.4509* | *0.0143* |
| | Distractor | –0.0088 | –0.0237, 0.0061 | 0.0076 | –1.1572 | 0.2472 |
| | THP | 0.0004 | –0.0153, 0.0145 | 0.0076 | 0.0490 | 0.9610 |
| | Incentive * distractor | –0.0042 | –0.0191, 0.0107 | 0.0076 | –0.5557 | 0.5785 |
| | Incentive * THP | –0.0002 | –0.0169, 0.0129 | 0.0076 | –0.2656 | 0.7906 |
| | Distractor * THP | –0.0095 | –0.0244, 0.0054 | 0.0076 | –1.2467 | 0.2125 |
| | Incentive * distractor * THP | 0.0054 | –0.0095, 0.0203 | 0.0076 | 0.7118 | 0.4766 |
| CNV (df = 1, 16,619) | *Incentive* | *–0.0917* | *–0.0106,–0.0771* | *0.0075* | *–12.258* | *<0.0001* |
| | Distractor | –0.0032 | –0.0178, 0.0115 | 0.0075 | –0.4278 | 0.6688 |
| | *THP* | *–0.0512* | *–0.0659,–0.0365* | *0.0075* | *–6.8409* | *<0.0001* |
| | Incentive * distractor | –0.0002 | –0.0149, 0.0144 | 0.0075 | –0.0301 | 0.9760 |
| | *Incentive * THP* | *0.0165* | *0.0018, 0.0311* | *0.0075* | *2.2036* | *0.0276* |
| | Distractor * THP | –0.0042 | –0.0188, 0.0105 | 0.0075 | –0.5560 | 0.5762 |
| | Incentive * distractor * THP | –0.0037 | –0.0184, 0.0109 | 0.0075 | –0.4974 | 0.6189 |
| Pre-preparation cue (df = 1, 15,879) | Incentive | –0.0006 | –0.0158, 0.0147 | 0.0078 | –0.0712 | 0.9430 |
| | Distractor | 0.0064 | –0.0089, 0.0216 | 0.0078 | 0.8186 | 0.4130 |
| | *THP* | *–0.0597* | *–0.0751,–0.0443* | *0.0078* | *–7.6126* | *<0.0001* |
| | Incentive * distractor | –0.0039 | –0.0191, 0.0114 | 0.0078 | –0.4959 | 0.6200 |
| | Incentive * THP | –0.0127 | –0.0279, 0.0026 | 0.0078 | –1.6256 | 0.1041 |
| | Distractor * THP | –0.0015 | –0.0168, 0.0137 | 0.0078 | –0.1963 | 0.8444 |
| | Incentive * distractor * THP | –0.0030 | –0.0183, 0.0123 | 0.0078 | –0.3843 | 0.7008 |

was strongly predicted by EEG voltage over a very large scalp area, centred on Cz from about 250 ms onwards. Distractor pull was predicted by frontal electrodes from about 120 ms after preparation cue, and this spread posteriorly to reach central and posterior electrodes by about 650 ms.

To check that these associations were not confounded by correlations between the saccadic measures themselves (e.g. RT is negatively correlated with residual velocity; r=–0.0681, p<0.0001), we re-ran this analysis while controlling for the other two saccadic measures. This did not materially change the results, indicating that preparatory EEG predicts these aspects of performance independently (*Figure 4—figure supplement 1*). An additional control analysis found that these results were not driven by microsaccades or ocular drift during the preparation period, as including these as trial-wise covariates did not substantially change the beta-coefficients (*Figure 4—figure supplement 2*).

## Preparatory activity explains incentive effects on RT but not movement speed

We have found that neural preparatory activity can predict residual velocity and RT, and is also affected by incentives and THP. Finally, we ask whether the neural activity can *explain* the effects of incentives and THP, through mediation analyses. We used the *Baron and Kenny, 1986*, method to assess mediation (see Methods/Analysis for full details). This tests whether the significant incentive effect on behaviour could be partially reduced (i.e. explained) by including the CNV as a mediator in a mixed-effects single-trial regression. We measured mediation as the reduction in (absolute)

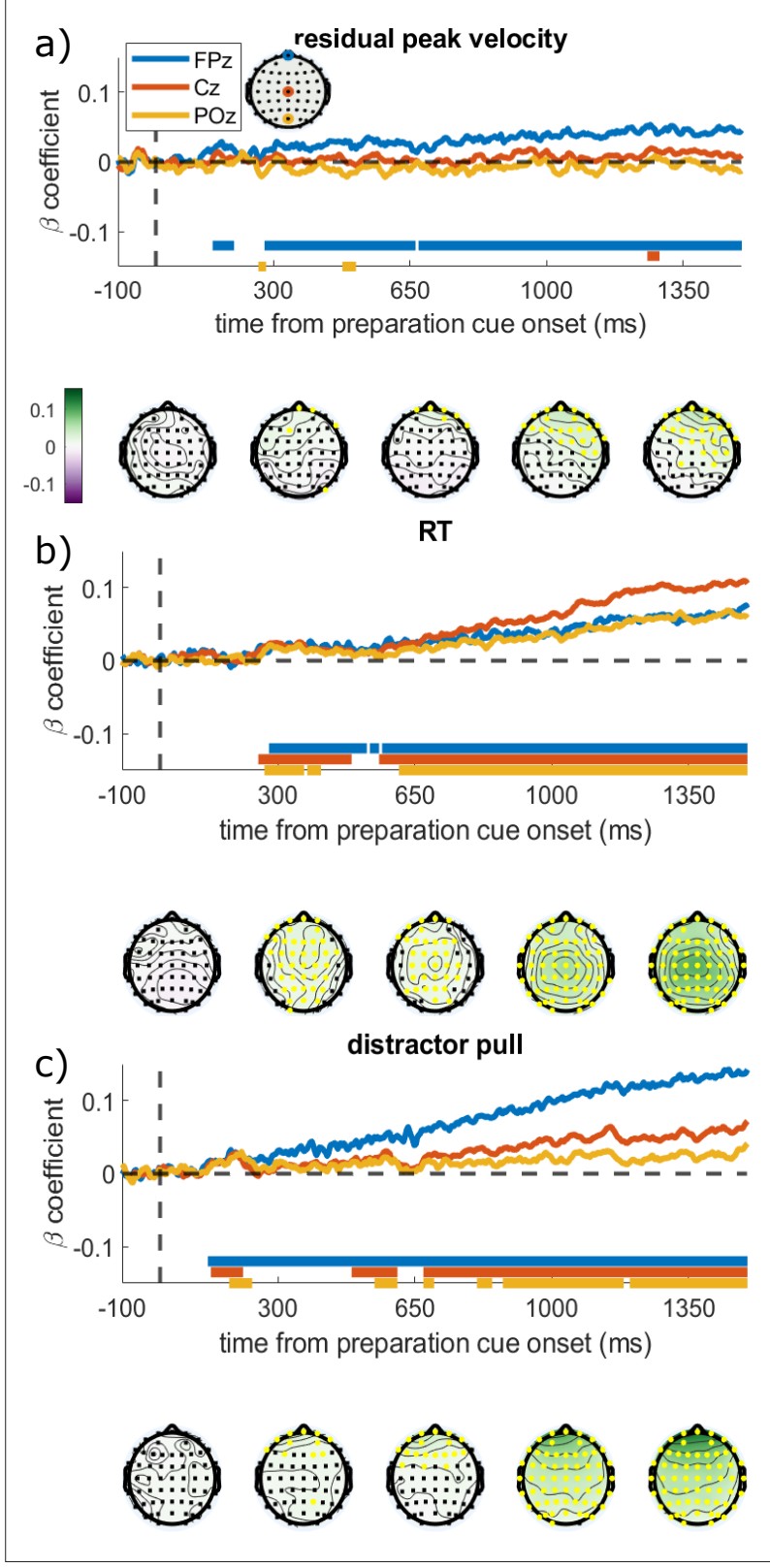

**Figure 4.** Regression coefficients from regressing each electrode and time-point against the different behavioural variables. The time series show the regression coefficients for three chosen electrodes, with the solid bars at the bottom showing significant clusters for those electrodes (family-wise error rate [FWER] = 0.05; 20 participants, 16627 trials). Topographies are shown below the graph at the times written on the x-axis, with the colours showing

*Figure 4 continued on next page*

*Figure 4 continued*

the regression coefficient and yellow electrodes showing significant clusters. (**a**) Residual velocity is predicted by voltage in the frontal electrodes briefly from 150:190 ms after preparation cue, and then again from 280 ms onwards, and gradually spreads backwards. (**b**) Reaction time (RT) is predicted by almost all electrodes from 250 ms onwards, and is strongest centrally. (**c**) Distractor pull is predicted by frontal electrodes from 120 ms and this grows stronger over time, the central and posterior electrodes have a brief cluster around 180 ms but only become consistently associated from 650 ms.

The online version of this article includes the following source data, source code, and figure supplement(s) for figure 4:

**Source code 1.** Matlab file to produce *Figure 4*.

**Source data 1.** mat file with data to produce *Figure 4*.

**Figure supplement 1.** Covariate-controlled regression coefficients from regressing each electrode and time-point against the different behavioural variables.

**Figure supplement 1—source data 1.** mat file with data to produce *Figure 4—figure supplement 1*.

**Figure supplement 2.** Regression analyses are relatively unchanged when controlling for eye movements.

**Figure supplement 2—source data 1.** mat file with data to produce *Figure 4—figure supplement 2*.

beta-coefficient for the incentive effect on behaviour when the CNV was included as a mediator (i.e. RT ~ 1 + incentive + CNV + incentive*CNV + (1 | participant)). This is a directional hypothesis of a reduced effect, and to assess significance we ran a permutation test, shuffling the CNV within participants, and measuring the change in absolute beta-coefficient for the incentive effect on behaviour. This generates a distribution of mediation effects where there is no relationship between CNV and RT on a trial (i.e. a null distribution). We ran 2500 permutations, and calculated the proportion with an equal or more negative change in absolute beta-coefficient, equivalent to a one-tailed test. We ran this mediation analysis separately for the two behavioural variables of RT and residual velocity, but not for distractor pull as it was not affected by incentive, so failed the assumptions of mediation analyses (*Baron and Kenny, 1986*; *Muller et al., 2005*). We took the mean CNV amplitude from 1200:1500 ms as our mediator.

Residual velocity passed all the assumption tests for mediation analysis, but no significant mediation was found (see *Figure 5*). That is, incentive predicted velocity ($\beta$=0.1304, $t(1,16,476)$=17.3280, $p<0.0001$); incentive predicted CNV ($\beta$=−0.9122, $t(1,16,476)$=−12.1800, $p<0.0001$); and CNV predicted velocity when included alongside incentive ($\beta$=0.0015, $t(1,16,475)$=1.9753, $p$=0.0483). However, including CNV did not reduce the incentive effect on velocity, and in fact strengthened it ($\beta$=0.1318, $t(1,16,475)$=17.4380, $p<0.0001$; change in absolute coefficient: $\Delta\beta$=+0.0014). Since there was no mediation (reduction), we did not run permutation tests on this.

However, RT did show a significant mediation of the incentive effect by CNV: incentive predicted RT: ($\beta$=−0.0868, $t(1,16,476)$=−14.9330, $p<0.0001$); incentive predicted CNV ($\beta$=−0.9122, $t(1,16,476)$=−12.1800, $p<0.0001$); and CNV predicted RT when included alongside incentive ($\beta$=0.0127, $t(1,16,475)$=21.3160, $p<0.0001$). The CNV mediated the effect of incentive on RT, reducing the absolute beta-coefficient ($\beta$=−0.0752, $t(1,16,475)$=−13.0570, $p<0.0001$; change in absolute coefficient: $\Delta\beta$=–0.0116). We assessed the significance of this change via permutation testing, shuffling the CNV across trials (within participants) and calculating the change in absolute beta-coefficient for the incentive effect on RT when the permuted CNV was included as a mediator. We repeated this 2500 times to build a null distribution of $\Delta\beta$, and calculated the proportion with equal or stronger reductions for a one-tailed $p$-value, which was highly significant ($p<0.0001$). This suggests that the incentive effect on RT is partially mediated by the CNV's amplitude during the preparation period, and this is not the case for residual velocity.

We also investigated whether the CNV could explain the cholinergic reduction in motivation (THP*incentive interaction) on RT – i.e., whether CNV mediated the THP moderation. We measured mediated moderation as suggested by *Muller et al., 2005*; see Methods/Analysis for full explanation: incentive*THP was associated with RT ($\beta$=0.0222, $t(1,16,474)$=3.8272, $p$=0.0001); and incentive*THP was associated with CNV ($\beta$=0.1619, $t(1,16,474)$=2.1671, $p$=0.0302); and CNV*THP was associated with RT ($\beta$=0.0014, $t(1,16,472)$=2.4061, $p$=0.0161). Mediated moderation was measured by the change in absolute incentive*THP effect when THP*CNV was included in the mixed-effects model

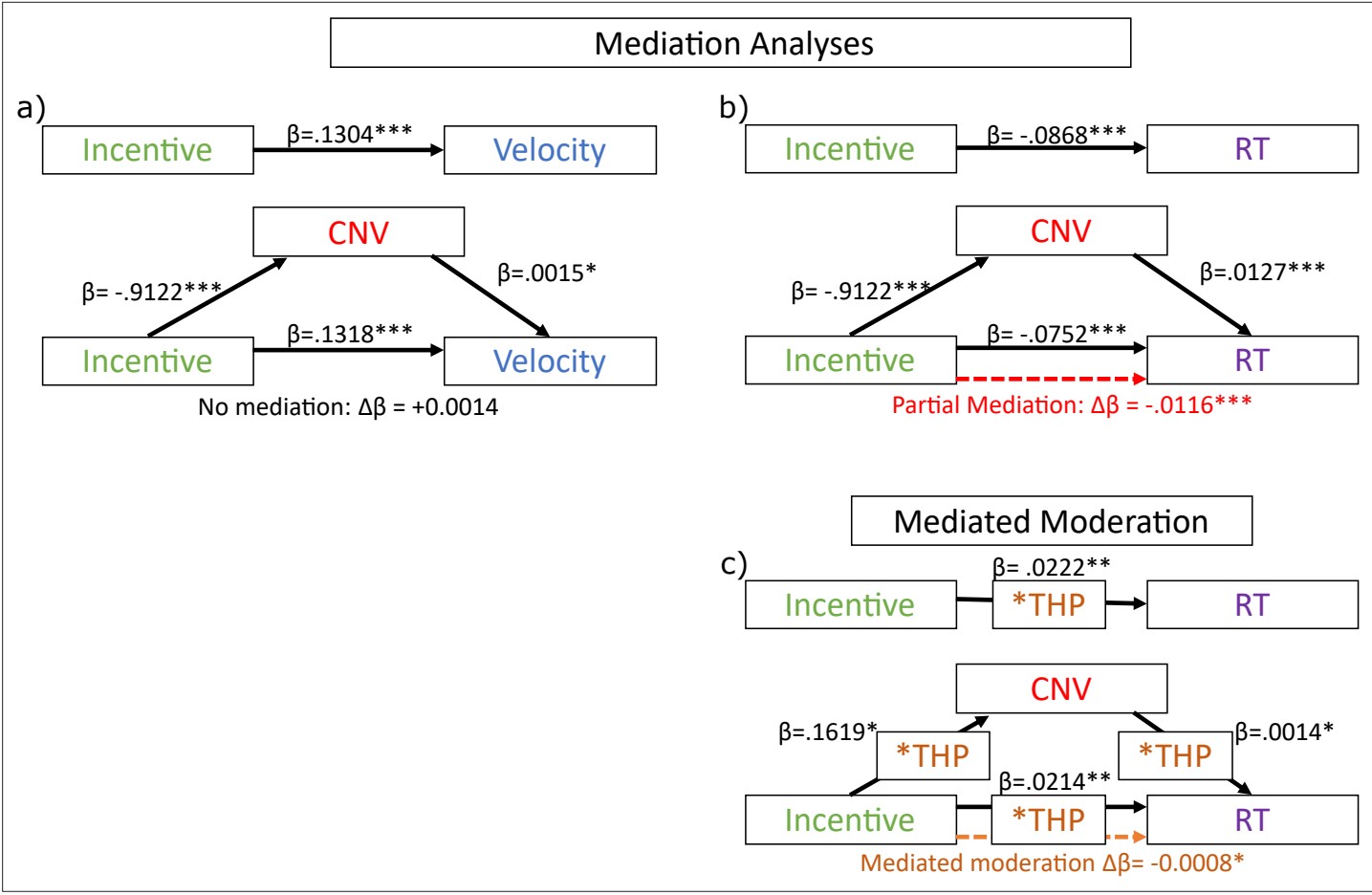

**Figure 5.** Mediation analyses of late contingent negative variation (CNV) amplitude on residual velocity and reaction times (RT). Black lines show significant associations, with single-trial regression coefficients (20 participants, 16627 trials), and significance indicated by the text, dashed red lines show significant mediations (permutation testing), and dashed orange lines show significant mediated moderations (permutation testing). (**a**) There was no mediation of the incentive effect on residual velocity by the mean CNV (1200:1500 ms). (**b**) However, CNV does partially mediate the effect of incentive on saccadic RT (dashed red line shows the indirect effect, which is the partial mediation). (**c**) Mediated moderation analysis: CNV mediated the moderation of trihexyphenidyl (THP) on the incentive effect.

The online version of this article includes the following source data for figure 5:

**Source data 1.** mat file with the full mediation analysis output for the residual velocity analysis.

**Source data 2.** mat file with the full mediation analysis output for the saccadic reaction time (RT) analysis.

($\beta$=0.0214, t(1,16,472)=3.7298, p=0.0002; change in beta-coefficient: $\Delta\beta$=–0.0008), and permutation testing (permuting the CNV as above) found a weakly significant effect (p=0.0132). This indicates cholinergic blockade changes how incentives affect preparatory negativity, and how this negativity reflects RT, which can explain some of the reduced invigoration of RT. However, this was not observed for saccade velocity.

## Discussion

When incentivised, participants initiated movements faster, and with faster velocity, but these motivational effects were reduced by blocking muscarinic acetylcholine receptors with THP (**Figure 1d and e**). THP also reduced repulsion away from a salient distractor (**Figure 2b**). The CNV, a fronto-central signal believed to reflect premotor preparatory activation, was stronger when incentives were present, and with THP (**Figure 3d**), and crucially, THP reduced the incentive benefit on CNV, mirroring the behavioural effects. Neural preparation predicted RT and distractibility (**Figure 3g**), with distinct scalp distributions (**Figure 4a–c**). The CNV partially mediated the incentivisation of RT

(*Figure 5b*), and could explain the drug-induced reduction in incentivisation of RT via a mediated moderation (*Figure 5c*). No such associations were seen in the window before the preparation cue onset, suggesting these effects relate specifically to preparing motivated action, rather than simply an incentive effect (*Figure 3b and e*). These results demonstrate a role of acetylcholine, and specifically muscarinic M1 receptors (M1rs), in motivation in humans.

Our behavioural measure of vigour was residual peak saccadic velocity, calculated separately within the drug and placebo sessions, and thus accounting for any overall drug effects on saccadic velocity or amplitude (or the linear relationship between the two, i.e. the main sequence). Previous studies have found this measure highly sensitive to incentivisation and motivation (*Grogan et al., 2020*; *Manohar et al., 2017*; *Muhammed et al., 2016*, *Muhammed et al., 2018*), with higher peak velocities than expected for a saccade of that amplitude, the same pattern seen here, and that was reduced when M1rs were antagonised. As antagonising M1rs reduced the incentivisation of residual peak velocity and saccadic RT, we can assume that M1rs normally play a facilitating role. This mirrors some studies using antimuscarinics in animal striatum to impair motivation (*Collins et al., 2016*; *Ostlund et al., 2014*; *Pratt and Kelley, 2004*), although those studies used scopolamine which antagonises M1-like and M2-like receptors, so effects may have been due to M2r antagonism. This complements previous work demonstrating that nicotine increases reward responsiveness (*Wang et al., 2020*) by enhancing striatal anticipatory activity. Basal ganglia outputs, along with cholinergic activity from the peduncu-lopontine nucleus, project to the superior colliculus to disinhibit saccade initiation (*Kobayashi and Isa, 2002*; *Naicker et al., 2017*) and also modulate ongoing instantaneous firing that controls velocity during saccades (*Smalianchuk et al., 2018*). This offers potential mechanisms for muscarinic receptors to influence movement vigour, together with other neurotransmitters such as dopamine, implicated in incentivisation (*Manohar et al., 2017*).

The three different ERPs used here reported differing effects: the incentive-cue response was affected by the drug but not by the incentive itself; the P3a was decreased by higher incentives but not affected by the drug; the CNV was the only one affected by both incentives and the drug, and showing an interaction of the two. The specificity to the CNV suggests that M1r antagonism is reducing incentivisation during the action-preparation process, and not affecting the overall sensitivity to the incentive. The CNV in the lead up to the target appearing was strengthened by the incentives, replicating previous work (*Frömer et al., 2021*; *Novak et al., 2016*; *Novak and Foti, 2015*; *Schever-nels et al., 2014*). M1r antagonism reduced the effect of incentives on CNV, mirroring the reduced incentivisation on RT and residual velocity. This suggests that cholinergic transmission is needed at an early stage of motivational preparation. CNV amplitude predicted RT, along with distractor pull, and frontal activity in this same time-window predicted residual velocity, aligning with recent studies linking CNV to RT and accuracy (*Frömer et al., 2021*), and to incentivisation of effort (*Berchio et al., 2019*). Our associations held up even when controlling for the other behavioural variables, suggesting they were not due to factors such as the negative correlation of RT and velocity. Together these find-ings suggest that incentives determine a baseline level of motivation, dependent on acetylcholine, that is integrated while we decide to act, thus influencing the speed of action initiation.

The CNV has been linked to activity in the supplementary motor area and cingulate cortex, depen-dent on inputs from ventral striatum and thalamus (*Nagai et al., 2004*; *Plichta et al., 2013*), and on a range of neurotransmitters including dopamine (*Linssen et al., 2011*) which also modulates moti-vational effects on vigour (*Grogan et al., 2020*; *Manohar and Husain, 2015*). Therefore a potential mechanism for our observed effects is that M1r activation in the striatum modulates the excitability of striatal pathways, and increases dopamine release as seen in animal work (*De Klippel et al., 1993*; *Galarraga et al., 1999*; *Shen et al., 2005*). Striatal muscarinic receptors could signal the pre-action, pre-decision level of motivation in humans.

No cognitive or computational model currently exists of how cholinergic modulation of dopamine affects aspects of motivation and motor control (*Grahek et al., 2020*). Previous studies have proposed that dopamine signals reward expectation which allows more cognitive and motor costs to be paid, in the form of both vigour and cognitive precision (*Ott and Nieder, 2019*; *Westbrook et al., 2021*). In contrast, acetylcholine has often been framed as increasing signal-to-noise ratio or amplifying atten-tion in cortical processing (*Bauer et al., 2012*; *Cragg, 2006*; *Laube et al., 2017*; *Sarter et al., 2006*). While a unified quantitative model will need considerable work (*Cools and Arnsten, 2022*), our data contribute by demonstrating that more than one locus of effect is likely.

The mediation analysis showed the incentivisation of RT could be partially explained by stronger CNV amplitudes, as could the antimuscarinic reduction in incentivisation. This mediated moderation was due to both reduced incentivisation of the CNV and reduced influence of the CNV on RT. These effects were rather small, which suggests the presence of an additional direct way for drug to affect incentivisation, perhaps via subcortical routes (*Faure et al., 2014*; *Mark et al., 2011*; *Mena-Segovia et al., 2008*) which could not be indexed by the ERPs in our study. There were no such mediations to explain the incentivisation of residual velocity, suggesting the CNV is not associated with the motivational preparation of *motor* speed of saccades. Animal studies link saccadic velocity to the instantaneous firing rate in the superior colliculus (*Smalianchuk et al., 2018*), which is inaccessible to EEG recordings, although the motivational modulation of this activity presumably arises from cortex or basal ganglia. Future investigations of other aspects of the EEG signals may illuminate us. Such studies could also investigate other potential signals that may be more sensitive to invigoration and/or muscarinic antagonism, including frequency-band power and phase coherence, or measures of variability in brain signals such as entropy, which may give greater insight into processes affected by these factors.

One potentially confusing finding is that although stronger CNV benefits RT (*Frömer et al., 2021*; *Novak and Foti, 2015*), M1r antagonism strengthened the CNV while *slowing* RT. The mediated moderation we found indicates that THP changes how CNV predicts RT, which suggests that THP has two effects: one upstream of the CNV strengthening anticipation, and one downstream, decreasing the coupling to RT. The former could be associated with ventral striatal drive to the CNV (*Plichta et al., 2013*), while the latter could reflect drug-induced changes in non-motivational cholinergic systems (*Gritton et al., 2016*).

While we have interpreted these effects as due to incentivisation, other closely related factors cannot be ruled out. When incentivised on this task, people expend more effort and also have a higher expectation of reward, which is linked to the effort they expend. Therefore, it is possible that the CNV is measuring the greater expected reward induced by motivation, which is linked to faster saccades (*Haith et al., 2012*; *Shadmehr et al., 2019*). The fact that no associations were seen between behaviour and neural activity in the time-window before the preparation cue might suggest that factors such as expected reward or arousal are less likely to explain our results. However, these explanations cannot be fully disentangled in this paradigm. We found that M1r antagonism led to a greater bias of distractor pull angles towards a salient distractor, which was unaffected by incentives. This fits with previous studies finding cholinergic involvement in attention and distraction (*Fallon et al., 2023*; *Gritton et al., 2016*; *Laube et al., 2017*; *Sarter et al., 2016*). However, we did *not* observe clear motivational improvements in distractibility, which have been seen previously (*Hickey and van Zoest, 2012*). This may be less surprising considering that the speed-accuracy trade-off dictates worse accuracy for faster saccades (*Reppert et al., 2018*) which is sensitive to the rewards available for responses (*Hickey and van Zoest, 2012*). Any motivational distraction effects might be weak in this study because the incentive schedule strongly favoured faster responses, and distractors were only present on half the trials, reducing the value of greater cognitive control. The repulsive bias away from the distractor was weakened by THP (*Figure 2d*), which may be a behavioural manifestation of reduced reactive top-down control reported with muscarinic antagonism (*Laube et al., 2017*). But in our data, the distractor pull could be predicted by preparatory activity over 1 s before the target onset, and this was also affected by muscarinic blockade. The frontal signature of distractor pull was distinct to the pattern predicting RT, suggesting that cholinergic effects on proactive control and speed are dissociable.

Care should be taken when interpreting a lack of effects in this study, given the reduced power due to a smaller than expected sample size. This may be more applicable to the whole-brain regressions and the mediation analysis, so future replications would be useful to verify these results. Additionally, as this study only tested young men, it is not known how generalisable the findings are to older adults, who may have age-induced changes in their cholinergic function, or indeed to patients with Parkinson's disease, who have additional dopaminergic deficits. Recent work showed that cholinesterase inhibitor withdrawal in patients with Parkinson's disease impaired a range of attentional and memory functions (*Fallon et al., 2023*), and investigating motivational impairments would be useful too. The effects of acetylcholine on motivation are of crucial importance in Parkinson's disease, where the balance between dopamine and acetylcholine is disrupted (*Pisani et al., 2007*; *Schulz et al., 2018*), leading to apathy or impulsivity (*Devos et al., 2014*). As THP is often used to treat tremor in

Parkinson's disease, the finding that it impairs motivation suggests that it may worsen motivational symptoms in a population already troubled by apathy.

## Conclusion

Muscarinic M1r antagonism reduced the incentivisation of saccadic peak velocity and RT, suggesting that normally M1r activity is important for motivation. The incentives strengthened the CNV, a preparatory EEG component, and this mediated the incentivisation of RT and the reduction of this incentivisation by the drug, implicating the CNV as a potential marker of muscarinic invigoration.

## Methods

### Design

We used a randomised, counterbalanced, double-blinded, placebo-controlled trial of THP. Participants were tested twice (minimum 1 week separation), once on placebo, and once on THP, making this a within-subject study. Ethical approval was granted by the University of Oxford MSIDREC (R45265/ RE001).

### Drugs

Participants were administered 2 mg THP or 200 mg lactose pills, both encapsulated in a digestible shell, and labelled A and B by a colleague not involved in the study. Participants were randomised to receive either tablet A or B on their first session, and B or A on their second. The experimenters did not know whether A or B contained the drug until all data were collected and pre-processed. On arrival, we checked participants felt well, and administered the dose. Testing began 1.5–2 hr later.

The risks of THP in pregnancy are unknown, but the Summary Product of Characteristics states that it 'should not be used during pregnancy unless clearly necessary'. As this was a basic research study with no immediate clinical applications, there was no justification for any risk of administering the drug during pregnancy, so we only recruited male participants to keep this risk at zero.

### Participants

Our sample size calculations suggested 27 participants would detect a 0.5 effect size with 0.05 sensitivity and 0.8 power. We recruited 27 male participants (see Drugs section above), but due to the pandemic, we had to halt the study part-way, with 20 completed participants and 5 participants who had completed one session only. We only analysed the 20 completed participants, which achieved a post hoc power of 0.7. The mean age was 28.15 years (SD = 8.03 years).

Participants read the information sheet, and gave written informed consent. Participants were screened for contraindications (e.g. cardiovascular disease, hypo/hypertension, cardiac arrhythmia, stroke, kidney/liver disease, psychiatric conditions, gastrointestinal haemorrhage, glaucoma, epilepsy, lactose hypersensitivity, porphyria) for the drug and placebo, and 1-lead ECG was taken to check for prolonged QTc interval of over 480 ms. A medical doctor checked all this information before the participant was admitted to the study.

### Task

To measure invigoration of saccades by incentives, we adapted an incentivised saccade task (*Manohar and Husain, 2015*), where participants had to make speeded saccades to a low-salience target in exchange for money, while avoiding a high-salience distractor. The task was run in Matlab R2018b and Psychtoolbox-3 (*Kleiner et al., 2007*).

Participants saw three grey circles each with a black fixation cross (*Figure 1a*) on a black screen (11° apart), and had to fixate on the one that turned pink after 500 ms (+0–100 ms jitter) for 300 ms. An audio *incentive cue* was played 200 ms after fixation, 1100 ms duration, of a voice saying '50p maximum' or '0p maximum' to indicate the maximum money available on this trial. Fixation was checked again after this to ensure fixation (300 ms + 100 ms wait), and then the black fixation cross turned white. This was the *preparation cue*, which occurred 1500 ms before the target onset. Then, one of the other two circles dimmed, indicating it was the target, and on 50% of trials the other circle simultaneously brightened, as a salient distractor. Participants were rewarded a proportion of the incentive based on how quickly they looked at the target circle, with an adaptive reward rule that

had an exponential fall-off depending on the average RT in the previous up to 20 trials. This kept the rewards received roughly constant over the task. If participants looked at the distractor, they would need to make a corrective eye movement to the target, which would slow the time to reach the target, and result in less reward. Once gaze reached the target, a feedback sound was played if a medium (10–30p) or large (>30p) reward was obtained, while the value of reward earned flashed in the centre of the target circle for 800 ms. A 1100 ms (+0–100 ms jitter) rest period followed each trial, where participants were allowed to blink. The target on one trial became the fixation location for the next trial.

There were thus four trial types, with two maximum incentive levels (50p or 0p) and the presence or absence of a distractor. There were six of each trial in a block, giving 24 trials, and 20 blocks (480 trials in total; 120 per condition). Participants could pause as long as they liked between blocks, with a minimum 4 s break. The first trial of each block also had an extra 4 s rest period before the trial began to ensure participants were settled and ready.

Participants were given 24 practice trials during the screening visit and before the main task on each main visit. These included an extra fixation check during the 1500 ms delay between the preparation cue and target onset, and if they moved their eye more than 1° during this time, the trial was flagged for repetition and the experimenter asked them to maintain fixation better. This training improved fixation during piloting.

In addition to this task, two other tasks were performed that are not reported in this paper, including a working memory task and a reversal learning task. We asked participants to rate how they felt on a visual analogue scale before and after the tasks, and participants also performed a pro- and anti-saccade task, and measurement of the pupillary light reflex.

## Eye-tracking

We tracked participants' eyes with an EyeLink 2000 (SR Research) at 1000 Hz. Participants were seated 75 cm from the screen, with their head on a chin-rest and forehead-rest. Nine-point calibration was performed at the start of the task, and one-point validation was done at the start of each block, with re-calibrations if necessary. Stimuli were shown on an ASUS VG248Qe3D screen (53×30 cm, 1920×1080 pixels, 60 Hz).

Saccades were parsed with standard criteria (velocity >30°s$^{-1}$, acceleration >8000°s$^{-2}$). We took the first saccade over 1° in amplitude within 100–900 ms after the target onset, and calculated velocity with a sliding window of 4 ms, excluding segments faster than 3000°s$^{-1}$ or where eye-tracking was lost. Saccades with peak velocities outside 50–1600°s$^{-1}$ were excluded.

Saccadic velocity is correlated with the amplitude of the saccade, an effect known as the main sequence (*Bahill et al., 1975*), and saccade amplitude can also be affected by reward (*Grogan et al., 2020*). To remove the effect of amplitude on velocity, we regressed peak velocity against amplitude within each participant and drug session, combining across incentive level and distractor level, and took the residual peak velocity for each individual trial as our main measure (*Figure 1c*). This reflects the difference between the velocity measured and the velocity predicted by the main sequence, with positive values meaning faster than expected velocity. By calculating velocity residuals for each drug condition separately, this method removes any overall effects of the drug on velocity or amplitude, and instead measures individual trial variations within that drug condition. Velocity residuals have previously been shown to be most sensitive to reward manipulations of vigour (*Blundell et al., 2018*; *Grogan et al., 2020*; *Manohar et al., 2017*; *Manohar et al., 2019*).

Saccadic RT was taken as the time between target onset and the start of the saccade (as detected by EyeLink velocity and acceleration criteria; *Figure 1b*) in ms; we used log RT for the analyses but plot raw RT. Distractor pull was measured as the angular deviation of the eye from a straight line linking the fixation and target circles (*Figure 2a*) when it left the fixation circle; positive values reflected a bias towards the distractor, while negative values reflected a bias away from the distractor.

## EEG acquisition and pre-processing

We recorded EEG with a Refa72 amplifier (TMSi, B.v Netherlands) at 1024 Hz and using OpenVibe software (*Renard et al., 2010*). We used a 64-channel cap (TMSi). The ground was placed on the left clavicle, and we recorded horizontal EOG with bipolar electrodes placed either side of the eyes. Due to the cap, we could not place an EOG electrode above the eye, so one was placed 1 cm under the

left eye, and this was converted into a bipolar EOG signal as the difference from electrode FP1, which was the closest cap electrode to the left eye. Impedances were kept below 10 kΩ as it was a high-impedance system.

Data were processed with custom Matlab scripts, and EEGLab and ERPLab toolboxes (*Delorme and Makeig, 2004*; *Lopez-Calderon and Luck, 2014*). Channels were referenced to the average of the two mastoid electrodes A1+A2, and synchronised with the eye-tracking traces using the EYE-EEG toolbox (*Dimigen et al., 2011*). Data were band-pass filtered at 0.1–80 Hz with an IIR Butterworth filter, notch filtered at 50 Hz with a stop-band Parks-McClellan filter, and down-sampled to 256 Hz.

Epochs were from –200:1500 ms around the preparation cue onset, and were baselined to the 100 ms before the preparation cue appeared. Visual inspection found no channels with outlying variance, so no channel rejection or interpolation was performed. We rejected trials from the EEG analyses where participants blinked or made saccades (according to EyeLink criteria above) during the epoch, or where EEG voltage in any channel was outside –200:200 μV (muscle activity). On average 104/120 trials per condition per person were included (SD = 21, range = 21–120), and 831/960 trials in total per person (SD = 160, range = 313–954). A repeated-measures ANOVA found there were no significant differences in number of trials excluded for any condition (p>0.2).

We took the late CNV period (1200:1500 ms) at electrode Cz as our a priori region of interest, along with the cue-P3 (200–280 ms). In order to see whether our results were specific to preparatory activity, we looked at activity before the preparation cue began, and looked at the late ERP to the incentive cue. We epoched the data from –200:1100 ms around the incentive cue onset (which was the duration of the incentive cue), and used the same artefact rejection criteria as above.

## Analysis

Behavioural and EEG analysis included all 20 participants, although trials with EEG artefacts were included in the behavioural analyses (18,585 trials in total) and not the EEG analyses (16,627 trials in total), to increase power in the former. Removing these trials did not change the findings of the behavioural analyses.

We used single-trial linear-mixed effects models to analyse our data, including participant as a random effect of intercept, with the formula '~1 + incentive*distractor*THP + (1 | participant)', which includes all lower-order interactions and main effects. We z-scored all factors to give standardised beta-coefficients.

For the difference-wave cluster-based permutation tests (*Figure 3—figure supplement 2*), we used the DMGroppe Mass Univariate toolbox (*Groppe et al., 2011*), with 2500 permutations, to control the FWER at 0.05. This was used for looking at difference waves to test the effects of incentive, THP, and the incentive*THP interaction (using difference of difference waves), across all EEG electrodes.

We adapted this toolbox to also run cluster-based permutation regressions to examine the relationship between the behavioural variables and the voltages at all EEG electrodes at each time-point. On each iteration we shuffled the voltages across trials within each condition and person, and regressed it against the behavioural variable, with the model '~1 + voltage + incentive*distractorPresent*THP + (1 | participant)'. The voltage term measured the association between voltage and the behavioural variable, after controlling for effects of incentive*distractor*THP on behaviour. By shuffling the voltages, we removed the relationship to the behavioural variable, to build the null distribution of t-statistics across electrodes and time samples. We used the 'cluster mass' method (*Bullmore et al., 1999*; *Groppe et al., 2011*; *Maris and Oostenveld, 2007*) to build the null distribution, and calculated the p-value as the proportion of this distribution further from zero than the true t-statistics (two-tailed test). Given the relatively small sample size here, these whole-brain analyses should not be taken as definitive.

For the mediation analysis, we followed the four-step process (*Baron and Kenny, 1986*; *Muller et al., 2005*), which requires four tests be met for the outcome (behavioural variable, e.g. RT), mediator (ERP, e.g. CNV), and the treatment (incentive):

1. Outcome is significantly associated with the treatment (RT ~1 + incentive + (1 | participant))
2. Mediator is significantly associated with the treatment (ERP ~1 + incentive + (1 | participant))
3. Mediator is significantly associated with the outcome (RT ~1 + incentive +ERP + (1 | participant))
4. And the inclusion of the mediator reduces the association between the treatment and outcome (incentive effect from model #3)

The mediation was measured by the reduction in the absolute standardised beta-coefficient between incentive and behaviour when the ERP mediator was included (model #3 vs model #1 above). We used permutation testing to quantify the likelihood of finding these mediations under the null hypothesis, achieved by shuffling the ERP across trials (within each participant) to remove any link between the ERP and behaviour. We repeated this 2500 times to build a null distribution of the change in absolute beta-coefficients for the RT ~ incentive effect when this permuted mediator was included (model #3 vs model #1). We calculated a one-tailed p-value by finding the proportion of the null distribution that was equal or more negative than the true value (as mediation is a one-tailed prediction). For this mediation analysis, we only included trials with valid ERP measures, even for the models without the ERP included (e.g. model #1), to keep the trial numbers and degrees of freedom the same.

Mediated moderation (*Muller et al., 2005*) was used to see whether the effect of THP (the moderator) on behaviour is mediated by the ERP, with the following tests (after the previous mediation tests were already satisfied):

5. THP moderates the incentive effect, via a significant treatment*moderator interaction on the outcome (RT ~1 + incentive + THP + incentive*THP + (1 | participant)).
6. THP moderates the incentive effect on the mediator, via a treatment*moderator interaction on the outcome (ERP ~1 + incentive +THP + incentive*THP + (1 | participant)).
7. THP's moderation of the incentive effect is mediated by the ERP, via a reduction in the association of treatment*moderator on the outcome when the treatment*moderator interaction is included (RT ~1 + incentive + THP + incentive*THP + ERP + ERP*THP + (1 | participant)).

Mediated moderation is measured as the reduction in absolute beta-coefficients for 'RT ~ incentive*THP' between model #5 and #7, which captures how much of this interaction could be explained by including the mediator*moderator interaction (ERP*THP in model #7). We tested the significance of this with permutation testing as above, permuting the ERP across trials (within participants) 2500 times, and building a null distribution of the change in the absolute beta-coefficients for RT ~ incentive*THP between models #7 and #5. We calculated a one-tailed p-value from the proportion of these that were equal or more negative than the true change.

## Acknowledgements

The authors would like to thank Andrea Bocincova and Yongzhi Huang for help with EEG setup and analysis. The work was funded by a Medical Research Council Clinician Scientist Fellowship to SGM (MR/P00878X). MVS was funded by the Medical Research Council (MC_ST_U16042). MJG is funded by the Jon Moulton Charity Trust (Guernsey). The Refa72 was funded by the Academy of Medical Sciences, Starter Grant for Clinical lecturers (MJG). The scheme is generously supported by the Wellcome Trust, Medical Research Council, British Heart Foundation, Versus Arthritis, Diabetes UK, and British Thoracic Society (through the Helen and Andrew Douglas bequest).

## Additional information

### Funding

| Funder | Grant reference number | Author |
|---|---|---|
| Medical Research Council | MR/P00878X | John P Grogan<br>Sanjay G Manohar |
| Medical Research Council | MC_ST_U16042 | Maaike MH van Swieten |
| Jon Moulton Charity Trust | | Martin J Gillies |
| Academy of Medical Sciences | | Sanjay G Manohar |

The funders had no role in study design, data collection and interpretation, or the decision to submit the work for publication.

## Author contributions
John P Grogan, Conceptualization, Data curation, Software, Formal analysis, Validation, Investigation, Visualization, Methodology, Writing - original draft, Project administration, Writing – review and editing; Matthias Raemaekers, Maaike MH van Swieten, Investigation, Writing – review and editing; Alexander L Green, Martin J Gillies, Resources, Writing – review and editing; Sanjay G Manohar, Conceptualization, Software, Supervision, Funding acquisition, Methodology, Writing – review and editing

## Author ORCIDs
John P Grogan ⓘ https://orcid.org/0000-0002-0463-8904
Maaike MH van Swieten ⓘ https://orcid.org/0000-0002-7361-9467
Alexander L Green ⓘ https://orcid.org/0000-0002-7262-7297
Sanjay G Manohar ⓘ https://orcid.org/0000-0003-0735-4349

## Ethics
Participants read the information sheet, and gave written informed consent before starting the study. Ethical approval was granted by the University of Oxford MSIDREC (R45265/RE001).

Reviewer #2 (Public review): https://doi.org/10.7554/eLife.98922.3.sa1
Reviewer #3 (Public review): https://doi.org/10.7554/eLife.98922.3.sa2
Author response https://doi.org/10.7554/eLife.98922.3.sa3

# Additional files

## Supplementary files
• MDAR checklist

## Data availability
Anonymous data have been publicly shared on OSF, analysis code is publicly shared on GitHub (copy archived at https://doi.org/10.5281/zenodo.5141792), and mean-level data and code to produce the figures are included as source data and code files with the manuscript.

The following datasets were generated:

| Author(s) | Year | Dataset title | Dataset URL | Database and Identifier |
|---|---|---|---|---|
| Grogan JP, Raemaekers M, van Swieten MMH, Green AL, Gillies MJ, Manohar SG | 2021 | Muscarinic Motivation via CNV | https://osf.io/v5hkm/ | Open Science Framework, v5hkm |
| Grogan JP | 2024 | johnPGrogan/muscarinic_ motivation_CNV: VoR | https://doi.org/10. 5281/zenodo.5141792 | Zenodo, 10.5281/ zenodo.5141792 |

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
