## [Editor Report · eLife Assessment]

The authors have reported an **important** study in which they use a double-blind design to explore pharmacological manipulations in the context of a behavioral task. While the sample size is small, the use of varied methodology, including electrophysiology, behavior, and pharmacology, makes this manuscript particularly notable. Overall, the findings are **solid** and motivate future explanations into the relationships between acetylcholine and motivation.

---

## [Referee Report · Reviewer #2 (Public review)]

Summary:

This work by Grogan and colleagues aimed to translate animal studies showing that acetylcholine plays a role in motivation by modulating the effects of dopamine on motivation. They tested this hypothesis with a placebo-controlled pharmacological study administering a muscarinic antagonist (trihexyphenidyl; THP) to a sample of 20 adult men performing an incentivized saccade task while undergoing electroencephalography (EEG). They found that reward increased vigor and reduced reaction times (RTs) and, importantly, these reward effects were attenuated by trihexyphenidyl. High incentives increased preparatory EEG activity (contingent negative variation), and though THP also increased preparatory activity, it also reduced this reward effect on RTs.

Strengths:

The researchers address a timely and potentially clinically relevant question with a within-subject pharmacological intervention and a strong task design. The results highlight the importance of the interplay between dopamine and other neurotransmitter systems in reward sensitivity and even though no Parkinson's patients were included in this study, the results could have consequences for patients with motivational deficits and apathy if validated in the future.

Weaknesses:

The main weakness of the study is the small sample size (N=20) that unfortunately is limited to men only. Generalizability and replicability of the conclusions remain to be assessed in future research with a larger and more diverse sample size and potentially a clinically relevant population. The EEG results do not shape a concrete mechanism of action of the drug on reward sensitivity.

---

## [Referee Report · Reviewer #3 (Public review)]

Summary:

Grogan et al examine a role for muscarinic receptor activation in action vigor in a saccadic system. This work is motivated by a strong literature linking dopamine to vigor, and some animal studies suggesting that ACH might modulate these effects, and is important because patient populations with symptoms related to reduced vigor are prescribed muscarinic antagonists. The authors use a motivated saccade task with distractors to measure the speed and vigor of actions in humans under placebo or muscarinic antagonism. They show that muscarinic antagonism blunts the motivational effects of reward on both saccade velocity and RT, and also modulates the distractibility of participants, in particular by increasing the repulsion of saccades away from distractors. They show that preparatory EEG signals reflect both motivation and drug condition, and make a case that these EEG signals mediate the effects of the drug on behavior.

Strengths:

This manuscript addresses an interesting and timely question and does so using an impressive within subject pharmacological design and a task well designed to measure constructs of interest. The authors show clear causal evidence that ACH affects different metrics of saccade generation related to effort expenditure and their modulation by incentive manipulations. The authors link these behavioral effects to motor preparatory signatures, indexed with EEG, that relate to behavioral measures of interest and in at least one case statistically mediate the behavioral effects of ACH antagonism.

Weaknesses:

A primary weakness of this paper is the sample size - since only 20 participants completed the study. The authors address the sample size in several places and I completely understand the reason for the reduced sample size (study halt due to covid). Nonetheless, it is worth stating explicitly that this sample size is relatively small for the effect sizes typically observed in such studies highlighting the need for future confirmatory studies.

---

## [Author Response]

The following is the authors’ response to the current reviews.

**Public Reviews:**

**Reviewer #2 (Public review):**
Summary:This work by Grogan and colleagues aimed to translate animal studies showing that acetylcholine plays a role in motivation by modulating the effects of dopamine on motivation. They tested this hypothesis with a placebo-controlled pharmacological study administering a muscarinic antagonist (trihexyphenidyl; THP) to a sample of 20 adult men performing an incentivized saccade task while undergoing electroencephalography (EEG). They found that reward increased vigor and reduced reaction times (RTs) and, importantly, these reward effects were attenuated by trihexyphenidyl. High incentives increased preparatory EEG activity (contingent negative variation), and though THP also increased preparatory activity, it also reduced this reward effect on RTs.Strengths:The researchers address a timely and potentially clinically relevant question with a within-subject pharmacological intervention and a strong task design. The results highlight the importance of the interplay between dopamine and other neurotransmitter systems in reward sensitivity and even though no Parkinson's patients were included in this study, the results could have consequences for patients with motivational deficits and apathy if validated in the future.Weaknesses:The main weakness of the study is the small sample size (N=20) that unfortunately is limited to men only. Generalizability and replicability of the conclusions remain to be assessed in future research with a larger and more diverse sample size and potentially a clinically relevant population. The EEG results do not shape a concrete mechanism of action of the drug on reward sensitivity.

We thank the reviewer for their time and their assessment of this manuscript, and we appreciate their helpful comments on the previous version.

We agree that the sample size being smaller than planned due to the pandemic restrictions is a weakness for this study, and hope that future studies into cholinergic effects on motivation in humans will use larger sample sizes. They should also ensure women are not excluded from sample populations, which will become even more important if the research progresses to clinical populations.

**Reviewer #3 (Public review):**
Summary:Grogan et al examine a role for muscarinic receptor activation in action vigor in a saccadic system. This work is motivated by a strong literature linking dopamine to vigor, and some animal studies suggesting that ACH might modulate these effects, and is important because patient populations with symptoms related to reduced vigor are prescribed muscarinic antagonists. The authors use a motivated saccade task with distractors to measure the speed and vigor of actions in humans under placebo or muscarinic antagonism. They show that muscarinic antagonism blunts the motivational effects of reward on both saccade velocity and RT, and also modulates the distractibility of participants, in particular by increasing the repulsion of saccades away from distractors. They show that preparatory EEG signals reflect both motivation and drug condition, and make a case that these EEG signals mediate the effects of the drug on behavior.Strengths:This manuscript addresses an interesting and timely question and does so using an impressive within subject pharmacological design and a task well designed to measure constructs of interest. The authors show clear causal evidence that ACH affects different metrics of saccade generation related to effort expenditure and their modulation by incentive manipulations. The authors link these behavioral effects to motor preparatory signatures, indexed with EEG, that relate to behavioral measures of interest and in at least one case statistically mediate the behavioral effects of ACH antagonism.Weaknesses:A primary weakness of this paper is the sample size - since only 20 participants completed the study. The authors address the sample size in several places and I completely understand the reason for the reduced sample size (study halt due to covid). Nonetheless, it is worth stating explicitly that this sample size is relatively small for the effect sizes typically observed in such studies highlighting the need for future confirmatory studies.

We thank the reviewer for their time and their assessment of this manuscript, and we appreciate their helpful comments on the previous version.

We agree that the small sample size is a weakness of the study, and hope that future work into cholinergic modulation of motivation can involve larger samples to replicate and extend this work.

**Recommendations for the authors:**

**Reviewer #2 (Recommendations for the authors):**
Thank you for addressing my comments and clarifying the analysis sections. Women can be included in such studies by performing a pregnancy test before each test session, but I understand how this could have added to the pandemic limitations. Best of luck with your future work!

Thank you for your time in reviewing this paper, and your helpful comments.

**Reviewer #3 (Recommendations for the authors):**
The authors have done a great job at addressing my concerns and I think that the manuscript is now very solid. That said, I have one minor concern.

Thank you for your time in reviewing this paper, and your helpful comments.

For descriptions of mass univariate analyses and cluster correction, I am still a bit confused on exactly what terms were in the regression. In one place, the authors state:On each iteration we shuffled the voltages across trials within each condition and person, and regressed it against the behavioural variable, with the model 'variable ~1 + voltage + incentive*distractorPresent*THP + (1 | participant)'.I take this to mean that the regression model includes a voltage regressor and a three-way interaction term, along with participant level intercept terms.However, elsewhere, the authors state:"We regressed each electrode and time-point against the three behavioural variables separately, while controlling for effects of incentive, distractor, THP, the interactions of those factors, and a random effect of participant."I take this to mean that the regression model included regressors for incentive, distractorPresent, THP, along with their 2 and 3 way interactions. I think that this seems like the more reasonable model - but I just want to (1) verify that this is what the authors did and (2) encourage them to articulate this more clearly and consistently throughout.

We apologise for the lack of clarity about the whole-brain regression analyses.

We used Wilkinson notation for this formula, where ‘A*B’ denotes ‘A + B + A:B’, so all main effects and lower-order interactions terms were included in the regression, as your second interpretation says. The model written out in full would be:

'variable ~1 + voltage + incentive + distractorPresent + THP + incentive*distractorPresent + incentive*THP + distractorPresent*THP + incentive*distractorPresent*THP + (1 | participant)'

We will clarify this in the Version of Record.

The following is the authors’ response to the original reviews.

**Public Reviews:**

**Reviewer #1 (Public Review):**
Summary:The authors used a motivated saccade task with distractors to measure response vigor and reaction time (RT) in healthy human males under placebo or muscarinic antagonism. They also simultaneously recorded neural activity using EEG with event-related potential (ERP) focused analyses. This study provides evidence that the muscarinic antagonist Trihexyphenidyl (THP) modulates the motivational effects of reward on both saccade velocity and RT, and also increases the distractibility of participants. The study also examined the correlational relationships between reaction time and vigor and manipulations (THP, incentives) with components of the EEG-derived ERPs. While an interesting correlation structure emerged from the analyses relating the ERP biomarkers to behavior, it is unclear how these potentially epiphenomenal biomarkers relate to relevant underlying neurophysiology.Strengths:This study is a logical translational extension from preclinical findings of cholinergic modulation of motivation and vigor and the CNV biomarker to a normative human population, utilizing a placebo-controlled, double-blind approach.While framed in the context of Parkinson's disease where cholinergic medications can be used, the authors do a good job in the discussion describing the limitations in generalizing their findings obtained in a normative and non-age-matched cohort to an aged PD patient population.The exploratory analyses suggest alternative brain targets and/or ERP components that relate to the behavior and manipulations tested. These will need to be further validated in an adequately powered study. Once validated, the most relevant biomarkers could be assessed in a more clinically relevant population.Weaknesses:The relatively weak correlations between the main experimental outcomes provide unclear insight into the neural mechanisms by which the manipulations lead to behavioral manifestations outside the context of the ERP. It would have been interesting to evaluate how other quantifications of the EEG signal through time-frequency analyses relate to the behavioral outcomes and manipulations.The ERP correlations to relevant behavioral outcomes were not consistent across manipulations demonstrating they are not reliable biomarkers to behavior but do suggest that multiple underlying mechanisms can give rise to the same changes in the ERP-based biomarkers and lead to different behavioral outcomes.

We thank the reviewer for their review and their comments.

We agree that these ERPs may not be reliable biomarkers yet, given the many-to-one mapping we observed where incentives and THP antagonism both affected the CNV in different ways, and hope that future studies will help clarify the use and limitations of the CNV as a potential biomarker of invigoration.

Our original hypothesis was specifically about the CNV as an index of preparatory behaviour, but we plan to look at potential changes to frequency characteristics in future work. We have included this in the discussion of future investigations. (page 16, line 428):

“Future investigations of other aspects of the EEG signals may illuminate us. Such studies could also investigate other potential signals that may be more sensitive to invigoration and/or muscarinic antagonism, including frequency-band power and phase-coherence, or measures of variability in brain signals such as entropy, which may give greater insight into processes affected by these factors.”

**Reviewer #2 (Public Review):**
Summary:This work by Grogan and colleagues aimed to translate animal studies showing that acetylcholine plays a role in motivation by modulating the effects of dopamine on motivation. They tested this hypothesis with a placebo-controlled pharmacological study administering a muscarinic antagonist (trihexyphenidyl; THP) to a sample of 20 adult men performing an incentivized saccade task while undergoing electroengephalography (EEG). They found that reward increased vigor and reduced reaction times (RTs) and, importantly, these reward effects were attenuated by trihexyphenidyl. High incentives increased preparatory EEG activity (contingent negative variation), and though THP also increased preparatory activity, it also reduced this reward effect on RTs.Strengths:The researchers address a timely and potentially clinically relevant question with a within-subject pharmacological intervention and a strong task design. The results highlight the importance of the interplay between dopamine and other neurotransmitter systems in reward sensitivity and even though no Parkinson's patients were included in this study, the results could have consequences for patients with motivational deficits and apathy if validated in the future.Weaknesses:The main weakness of the study is the small sample size (N=20) that unfortunately is limited to men only. The generalizability and replicability of the conclusions remain to be assessed in future research with a larger and more diverse sample size and potentially a clinically relevant population. The EEG results do not shape a concrete mechanism of action of the drug on reward sensitivity.

We thank the reviewer for their review, and their comments.

We agree that our study was underpowered, not reaching our target of 27 participants due to pandemic restrictions halting our recruitment, and hope that future studies into muscarinic antagonism in motivation will have larger sample sizes, and include male and female participants across a range of ages, to assess generalisability.

We only included men to prevent the chance of administering the drug to someone pregnant. Trihexyphenidyl is categorized by the FDA as a Pregnancy Category Class C drug, and the ‘Summary of Product Characteristics’ states: “There is inadequate information regarding the use of trihexyphenidyl in pregnancy. Animal studies are insufficient with regard to effects on pregnancy, embryonal/foetal development, parturition and postnatal development. The potential risk for humans is unknown. Trihexyphenidyl should not be used during pregnancy unless clearly necessary.”

While the drug can be prescribed where benefits may outweigh this risk, as there were no benefits to participants in this study, we only recruited men to keep the risk at zero.

We have updated the Methods/Drugs section to explain this (page 17, line 494):

“The risks of Trihexyphenidyl in pregnancy are unknown, but the Summary Product of Characteristics states that it “should not be used during pregnancy unless clearly necessary”. As this was a basic research study with no immediate clinical applications, there was no justification for any risk of administering the drug during pregnancy, so we only recruited male participants to keep this risk at zero.”

And we reference to this in the Methods/Participants section (page 18, line 501):

“We recruited 27 male participants (see Drugs section above),…”

We agree that future work is needed to replicate this in different samples, and that this work cannot tell us the mechanism by which the drug is dampening invigoration, but we think that showing these effects do occur and can be linked to anticipatory/preparatory activity rather than overall reward sensitivity is a useful finding.

**Reviewer #3 (Public Review):**
Summary:Grogan et al examine a role for muscarinic receptor activation in action vigor in a saccadic system. This work is motivated by a strong literature linking dopamine to vigor, and some animal studies suggesting that ACH might modulate these effects, and is important because patient populations with symptoms related to reduced vigor are prescribed muscarinic antagonists. The authors use a motivated saccade task with distractors to measure the speed and vigor of actions in humans under placebo or muscarinic antagonism. They show that muscarinic antagonism blunts the motivational effects of reward on both saccade velocity and RT, and also modulates the distractibility of participants, in particular by increasing the repulsion of saccades away from distractors. They show that preparatory EEG signals reflect both motivation and drug condition, and make a case that these EEG signals mediate the effects of the drug on behavior.Strengths:This manuscript addresses an interesting and timely question and does so using an impressive within-subject pharmacological design and a task well-designed to measure constructs of interest. The authors show clear causal evidence that ACH affects different metrics of saccade generation related to effort expenditure and their modulation by incentive manipulations. The authors link these behavioral effects to motor preparatory signatures, indexed with EEG, that relate to behavioral measures of interest and in at least one case statistically mediate the behavioral effects of ACH antagonism.Weaknesses:In full disclosure, I have previously reviewed this manuscript in another journal and the authors have done a considerable amount of work to address my previous concerns. However, I have a few remaining concerns that affect my interpretation of the current manuscript.Some of the EEG signals (figures 4A&C) have profiles that look like they could have ocular, rather than central nervous, origins. Given that this is an eye movement task, it would be useful if the authors could provide some evidence that these signals are truly related to brain activity and not driven by ocular muscles, either in response to explicit motor effects (ie. Blinks) or in preparation for an upcoming saccade.

We thank the reviewer for re-reviewing the manuscript and for raising this issue.

All the EEG analyses (both ERP and whole-brain) are analysing the preparation period between the ready-cue and target appearance when no eye-movements are required. We reject trials with blinks or saccades over 1 degree in size, as detected by the Eyelink software according the sensitive velocity and acceleration criteria specified in the manuscript (Methods/Eye-tracking, page 19, line 550). This means that there should be no overt eye movements in the data. However, microsaccades and ocular drift are still possible within this period, which indeed could drive some effects. To measure this, we counted the number of microsaccades (<1 degree in size) in the preparation period between incentive cue and the target onset, for each trial. Further, we measure the mean absolute speed of the eye during the preparation period (excluding the periods during microsaccades) for each trial.

We have run a control analysis to check whether including ocular drift speed or number of microsaccades as a covariate in the whole-brain regression analysis changes the association between EEG and the behavioural metrics at frontal or other electrodes. Below we show these ‘variable ~ EEG’ beta-coefficients when controlling for each eye-movement covariate, in the same format as Figure 4. We did not run the permutation testing on this due to time/computational costs (it takes >1 week per variable), so p-values were not calculated, only the beta-coefficients. The beta-coefficients are almost unchanged, both in time-course and topography, when controlling for either covariate. The frontal associations to velocity and distractor pull remain, suggesting they are not due to these eye movements.

We have added this figure as a supplemental figure.

For additional clarity in this response, we also plot the differences between these covariate-controlled beta-coefficients, and the true beta-coefficients from figure 4 (please note the y-axis scales are -0.02:0.02, not -0.15:0.15 as in Figure 4 and Figure 4-figure supplement 2). This shows that the changes to the associations between EEG and velocity/distractor-pull were not frontally-distributed, demonstrating eye-movements were not driving these effects. Relatedly, the RT effect’s change was frontally-distributed, despite Figure 4 showing the true relationship was central in focus, again indicating that effect was also not related to these eye movements.

**Author response image 1. sa3fig1:** Difference in beta-coefficients when eye-movement covariates are included. This is the difference from the beta-coefficients shown in Figure 4, please note the smaller y-axis limits.

The same pattern was seen if we controlled for the change in eye-position from the baseline period (measured by the eye-tracker) at each specific time-point, i.e., controlling for the distance the eye had moved from baseline at the time the EEG voltage is measured. The topographies and time-course plots were almost identical to the above ones:

**Author response image 2. sa3fig2:** Controlling for change in eye-position at each time-point does not change the regression results. Left column shows the beta-coefficients between the variable and EEG voltage, and the right column shows the difference from the main results in Figure 4 (note the smaller y-axis limits for the right-hand column).

Therefore, we believe the brain-behaviour regressions are independent of eye-movements. We have included the first figure presented here as an additional supplemental figure, and added the following to the text (page 10, line 265):

“An additional control analysis found that these results were not driven by microsaccades or ocular drift during the preparation period, as including these as trial-wise covariates did not substantially change the beta-coefficients (Figure 4 – Figure Supplement 2).”

For other EEG signals, in particular, the ones reported in Figure 3, it would be nice to see what the spatial profiles actually look like - does the scalp topography match that expected for the signal of interest?

Yes, the CNV is a central negative potential peaking around Cz, while the P3a is slightly anterior to this (peaking between Cz and FCz). We have added the topographies to the main figure (see point below).

This is the topography of the mean CNV (1200:1500ms from the preparation cue onset), which is maximal over Cz, as expected.

The P3a’s topography (200:280ms after preparation cue) is maximal slightly anterior to Cz, between Cz and FCz.

A primary weakness of this paper is the sample size - since only 20 participants completed the study. The authors address the sample size in several places and I completely understand the reason for the reduced sample size (study halt due to COVID). That said, they only report the sample size in one place in the methods rather than through degrees of freedom in their statistical tests conducted throughout the results. In part because of this, I am not totally clear on whether the sample size for each analysis is the same - or whether participants were removed for specific analyses (ie. due to poor EEG recordings, for example).

We apologise for the lack of clarity here. All 20 participants were included in all analyses, although the number of trials included differed between behavioural and EEG analyses. We only excluded trials with EEG artefacts from the EEG analyses, not from the purely behavioural analyses such as Figures 1&2, although trials with blinks/saccades were removed from behavioural analyses too. Removing the EEG artefactual trials from the behavioural analyses did not change the findings, despite the lower power. The degrees of freedom in the figure supplement tables are the total number of trials (less 8 fixed-effect terms) included in the single-trial / trial-wise regression analyses we used.

We have clarified this in the Methods/Analysis (page 20, line 602):

“Behavioural and EEG analysis included all 20 participants, although trials with EEG artefacts were included in the behavioural analyses (18585 trials in total) and not the EEG analyses (16627 trials in total), to increase power in the former. Removing these trials did not change the findings of the behavioural analyses.”

And we state the number of participants and trials in the start of the behavioural results (page 3, line 97):

“We used single-trial mixed-effects linear regression (20 participants, 18585 trials in total) to assess the effects of Incentive, Distractors, and THP, along with all the interactions of these (and a random-intercept per participant), on residual velocity and saccadic RT.”

and EEG results section (page 7, line 193):

“We used single-trial linear mixed-effects regression to see the effects of Incentive and THP on each ERP (20 participants, 16627 trials; Distractor was included too, along with all interactions, and a random intercept by participant).”

Beyond this point, but still related to the sample size, in some cases I worry that results are driven by a single subject. In particular, the interaction effect observed in Figure 1e seems like it would be highly sensitive to the single subject who shows a reverse incentive effect in the drug condition.

Repeating that analysis after removing the participant with the large increase in saccadic RT with incentives did not remove the incentive*THP interaction effect – although it did weaken slightly from (β = 0.0218, p = .0002) to (β=0.0197, p=.0082). This is likely because that while that participant did have slower RTs for higher incentives on THP, they were also slower for higher incentives under placebo (and similarly for distractor present/absent), making them less of an outlier in terms of effects than in raw RT terms. Below is Author response image 3 the mean-figure without that participant, and Author response image 4 that participant shown separately.

**Author response image 3. sa3fig3:** 

**Author response image 4. sa3fig4:** 

There are not sufficient details on the cluster-based permutation testing to understand what the authors did or whether it is reasonable. What channels were included? What metric was computed per cluster? How was null distribution generated?

We apologise for not giving sufficient details of this, and have updated the Methods/Analysis section to include these details, along with a brief description in the Results section.

To clarify here, we adapted the DMGroppe Mass Univariate Testing toolbox to also run cluster-based permutation regressions to examine the relationship between the behavioural variables and the voltages at all EEG electrodes at each time point. On each iteration we shuffled the voltages across trials within each condition and person, and regressed it against the behavioural variable, with the model ‘variable ~1 + voltage + incentive*distractorPresent*THP + (1 | participant)’. The Voltage term measured the association between voltage and the behavioural variable, after controlling for effects of incentive*distractor*THP on behaviour – i.e. does adding the voltage at this time/channel explain additional variance in the variable not captured in our main behavioural analyses. By shuffling the voltages, we removed the relationship to the behavioural variable, to build the null distribution of t-statistics across electrodes and time-samples. We used the ‘cluster mass’ method (Bullmore et al., 1999; Groppe et al., 2011; Maris & Oostenveld, 2007) to build the null distribution of cluster mass (across times/channels per iteration), and calculated the p-value as the proportion of this distribution further from zero than the absolute true t-statistics (two-tailed test).

We have given greater detail for this in the Methods/Analysis section (page 20, line 614):

“We adapted this toolbox to also run cluster-based permutation regressions to examine the relationship between the behavioural variables and the voltages at all EEG electrodes at each time point. On each iteration we shuffled the voltages across trials within each condition and person, and regressed it against the behavioural variable, with the model ‘~1 + voltage + incentive*distractorPresent*THP + (1 | participant)’. The Voltage term measured the association between voltage and the behavioural variable, after controlling for effects of incentive*distractor*THP on behaviour. By shuffling the voltages, we removed the relationship to the behavioural variable, to build the null distribution of t-statistics across electrodes and time-samples. We used the ‘cluster mass’ method (Bullmore et al., 1999; Groppe et al., 2011; Maris & Oostenveld, 2007) to build the null distribution, and calculated the p-value as the proportion of this distribution further from zero than the true t-statistics (two-tailed test). Given the relatively small sample size here, these whole-brain analyses should not be taken as definitive.”

And we have added a brief explanation to the Results section also (page 9, line 246):

“We regressed each electrode and time-point against the three behavioural variables separately, while controlling for effects of incentive, distractor, THP, the interactions of those factors, and a random effect of participant. This analysis therefore asks whether trial-to-trial neural variability predicts behavioural variability. To assess significance, we used cluster-based permutation tests (DMGroppe Mass Univariate toolbox; Groppe, Urbach, & Kutas, 2011), shuffling the trials within each condition and person, and repeating it 2500 times, to build a null distribution of ‘cluster mass’ from the t-statistics (Bullmore et al., 1999; Maris & Oostenveld, 2007) which was used to calculate two-tailed p-values with a family-wise error rate (FWER) of .05 (see Methods/Analysis for details).”

The authors report that "muscarinic antagonism strengthened the P3a" - but I was unable to see this in the data plots. Perhaps it is because the variability related to individual differences obscures the conditional differences in the plots. In this case, event-related difference signals could be helpful to clarify the results.

We thank the reviewer for spotting this wording error, this should refer to the *incentive* effect weakening the P3a, as no other significant effects were found on the P3a, as stated correctly in the previous paragraph. We have corrected this in the manuscript (page 9, line 232):

“This suggests that while incentives strengthened the incentive-cue response and the CNV and weakened the P3a, muscarinic antagonism strengthened the CNV,”

The reviewer’s suggestion for difference plots is very valuable, and we have added these to Figure 3, as well as increasing the y-axis scale for figure 3c to show the incentives weakening the P3a more clearly, and adding the topographies suggested in an earlier comment. The difference waves for Incentive and THP effects show that both are decreasing voltage, albeit with slightly different onset times – Incentive starts earlier, thus weakening the positive P3a, while both strengthen the negative CNV. The Incentive effects within THP and Placebo separately illustrate the THP*Incentive interaction.

We have amended the Results text and figure (page 7, line 200):

“The subsequent CNV was strengthened (i.e. more negative; Figure 3d) by incentive (β = -.0928, p < .0001) and THP (β = -0.0502, p < .0001), with an interaction whereby *THP decreased the incentive effect* (β = 0.0172, p = .0213). Figure 3h shows the effects of Incentive and THP on the CNV separately, using difference waves, and Figure 3i shows the incentive effect grows more slowly in the THP condition than the Placebo condition.

For mediation analyses, it would be useful in the results section to have a much more detailed description of the regression results, rather than just reporting things in a binary did/did not mediate sort of way. Furthermore, the methods should also describe how mediation was tested statistically (ie. What is the null distribution that the difference in coefficients with/without moderator is tested against?).

We have added a more detailed explanation of how we investigated mediation and mediated moderation, and now report the mediation effects for all tests run and the permutation-test p-values.

We had been using the Baron & Kenny (1986) method, based on 4 tests outlined in the updated text below, which gives a single measure of change in absolute beta-coefficients when all the tests have been met, but without any indication of significance; any reduction found after meeting the other 3 tests indicates a partial mediation under this method. We now use permutation testing to generate a p-value for the likelihood of finding an equal or larger reduction in the absolute beta-coefficients if the CNV were not truly related to RT. This found that the CNV’s mediation of the Incentive effect on RT was highly significant, while the Mediated Moderation of CNV on THP*Incentive was weakly significant.

During this re-analysis, we noticed that we had different trial-numbers in the different regression models, as EEG-artefactual trials were not excluded from the behavioural-only model (‘RT ~ 1 + Incentive’). However, this causes issues with the permutation testing as we are shuffling the ERPs and need the same trials included in all the mixed-effects models. Therefore, we have redone these mediation analyses, including only the trials with valid ERP measures (i.e. no artefactual trials) in all models. This has changed the beta-coefficients we report, but not the findings or conclusions of the mediation analyses. We have updated the figure to have these new statistics.

We have updated the text to explain the methodology in the Results section (page 12, line 284):

“We have found that neural preparatory activity can predict residual velocity and RT, and is also affected by incentives and THP. Finally, we ask whether the neural activity can *explain* the effects of incentives and THP, through mediation analyses. We used the Baron & Kenny (1986) method to assess mediation (see Methods/Analysis for full details). This tests whether the significant Incentive effect on behaviour could be partially reduced (i.e., explained) by including the CNV as a mediator in a mixed-effects single-trial regression. We measured mediation as the reduction in (absolute) beta-coefficient for the incentive effect on behaviour when the CNV was included as a mediator (i.e., RT ~ 1 + Incentive + CNV + Incentive*CNV + (1 | participant)). This is a directional hypothesis of a reduced effect, and to assess significance we ran a permutation-test, shuffling the CNV within participants, and measuring the change in absolute beta-coefficient for the Incentive effect on behaviour. This generates a distribution of mediation effects where there is no relationship between CNV and RT on a trial (i.e., a null distribution). We ran 2500 permutations, and calculated the proportion with an equal or more negative change in absolute beta-coefficient, equivalent to a one-tailed test. We ran this mediation analysis separately for the two behavioural variables of RT and residual velocity, but not for distractor pull as it was not affected by incentive, so failed the assumptions of mediation analyses (Baron & Kenny, 1986; Muller et al., 2005). We took the mean CNV amplitude from 1200:1500ms as our Mediator.

Residual velocity passed all the assumption tests for Mediation analysis, but no significant mediation was found. That is, Incentive predicted velocity (β=0.1304, t(1,16476)=17.3280, p<.0001); Incentive predicted CNV (β=-0.9122, t(1,16476)=-12.1800, p<.0001); and CNV predicted velocity when included alongside Incentive (β=0.0015, t(1,16475)=1.9753, p=.0483). However, including CNV did not reduce the Incentive effect on velocity, and in fact strengthened it (β=0.1318, t(1,16475)=17.4380, p<.0001; change in absolute coefficient: Δβ=+0.0014). Since there was no mediation (reduction), we did not run permutation tests on this.

However, RT did show a significant mediation of the Incentive effect by CNV: Incentive predicted RT (β=-0.0868, t(1,16476)=-14.9330, p<.0001); Incentive predicted CNV (β=-0.9122, t(1,16476)=-12.1800, p<.0001); and CNV predicted RT when included alongside Incentive (β=0.0127, t(1,16475)=21.3160, p<.0001). The CNV mediated the effect of Incentive on RT, reducing the absolute beta-coefficient (β=-0.0752, t(1,16475)=-13.0570, p<.0001; change in absolute coefficient: Δβ = -0.0116). We assessed the significance of this change via permutation testing, shuffling the CNV across trials (within participants) and calculating the change in absolute beta-coefficient for the Incentive effect on RT when the permuted CNV was included as a mediator. We repeated this 2500 times to build a null distribution of Δβ, and calculated the proportion with equal or stronger reductions for a one-tailed p-value, which was highly significant (p<.0001). This suggests that the Incentive effect on RT is partially mediated by the CNV’s amplitude during the preparation period, and this is not the case for residual velocity.

We also investigated whether the CNV could explain the cholinergic reduction in motivation (THP*Incentive interaction) on RT – i.e., whether CNV mediation the THP moderation. We measured Mediated Moderation as suggested by Muller et al. (2005; see Methods/Analysis for full explanation): Incentive*THP was associated with RT (β=0.0222, t(1,16474)=3.8272, p=.0001); and Incentive*THP was associated with CNV (β=0.1619, t(1,16474)=2.1671, p=.0302); and CNV*THP was associated with RT (β=0.0014, t(1,16472)=2.4061, p=.0161). Mediated Moderation was measured by the change in absolute Incentive*THP effect when THP*CNV was included in the mixed-effects model (β=0.0214, t(1,16472)=3.7298, p=.0002; change in beta-coefficient: Δβ = -0.0008), and permutation-testing (permuting the CNV as above) found a significant effect (p=.0132). This indicates cholinergic blockade changes how incentives affect preparatory negativity, and how this negativity reflects RT, which can explain some of the reduced invigoration of RT. However, this was not observed for saccade velocity.

And we have updated the Methods/Analysis section with a more detailed explanation too (page 21, line 627):

“For the mediation analysis, we followed the 4-step process (Baron & Kenny, 1986; Muller et al., 2005), which requires 4 tests be met for the outcome (behavioural variable, e.g. RT), mediator (ERP, e.g., CNV) and the treatment (Incentive):

(1) Outcome is significantly associated with the Treatment (RT ~ 1 + Incentive + (1 | participant))(2) Mediator is significantly associated with the Treatment (ERP ~ 1 + Incentive + (1 | participant))(3) Mediator is significantly associated with the Outcome (RT ~ 1 + Incentive + ERP + (1 | participant))(4) And the inclusion of the Mediator reduces the association between the Treatment and Outcome (Incentive effect from model #3)

The mediation was measured by the reduction in the absolute standardised beta coefficient between incentive and behaviour when the ERP mediator was included (model #3 vs model #1 above). We used permutation-testing to quantify the likelihood of finding these mediations under the null hypothesis, achieved by shuffling the ERP across trials (within each participant) to remove any link between the ERP and behaviour. We repeated this 2500 times to build a null distribution of the change in absolute beta-coefficients for the RT ~ Incentive effect when this permuted mediator was included (model #3 vs model #1). We calculated a one-tailed p-value by finding the proportion of the null distribution that was equal or smaller than the true values (as Mediation is a one-tailed prediction).

Mediated moderation (Muller et al., 2005) was used to see whether the effect of THP (the Moderator) on behaviour is mediated by the ERP, with the following tests (after the previous Mediation tests were already satisfied):

(5) THP moderates the Incentive effect, via a significant Treatment*Moderator interaction on the Outcome (RT ~ 1 + Incentive + THP + Incentive*THP + (1 | participant))(6) THP moderates the Incentive effect on the Mediator, via a Treatment*Moderator interaction on the Outcome (ERP ~ 1 + Incentive + THP + Incentive*THP + (1 | participant))(7) THP’s moderation of the Incentive effect is mediated by the ERP, via a reduction in the association of Treatment*Moderator on the Outcome when the Treatment*Moderator interaction is included (RT ~ 1 + Incentive + THP + Incentive*THP + ERP + ERP*THP + (1 | participant))

Mediated moderation is measured as the reduction in absolute beta-coefficients for ‘RT ~ Incentive*THP’ between model #5 and #7, which captures how much of this interaction could be explained by including the Mediator*Moderator interaction (ERP*THP in model #7). We tested the significance of this with permutation testing as above, permuting the ERP across trials (within participants) 2500 times, and building a null distribution of the change in the absolute beta-coefficients for RT ~ Incentive*THP between models #7 and #5. We calculated a one-tailed p-value from the proportion of these that were equal or smaller than the true change.”
**Recommendations for the authors:**

**Reviewer #2 (Recommendations For The Authors):**
(1) The analysis section could benefit from greater detail. For example, how exactly did they assess that the effects of the drug on peak velocity and RT were driven by non-distracting trials? Ideally, for every outcome, the analysis approach used should be detailed and justified.

We apologise for the confusion from this. To clarify, we found a 2-way regression (incentive*THP) on both residual velocity and saccadic RT and this pattern was stronger in distractor-absent trials for residual velocity, and stronger in distractor-present trials for saccadic RT, as can be seen in Figure 1d&e. However, as there was no significant 3-way interaction (incentive*THP*distractor) for either metric, and the 2-way interaction effects were in the same direction in distractor present/absent trials for both metrics, we think these effects were relatively unaffected by distractor presence.

We have updated the Results section to make this clearer: (page 3, line 94):

We measured vigour as the residual peak velocity of saccades within each drug session (see Figure 1c & Methods/Eye-tracking), which is each trial’s deviation of velocity from the main sequence. This removes any overall effects of the drug on saccade velocity, while still allowing incentives and distractors to have different effects within each drug condition. We used single-trial mixed-effects linear regression (20 participants, 18585 trials in total) to assess the effects of Incentive, Distractors, and THP, along with all the interactions of these (and a random-intercept per participant), on residual velocity and saccadic RT. As predicted, residual peak velocity was increased by incentives (Figure 1d; β = 0.1266, p < .0001), while distractors slightly slowed residual velocity (β = -0.0158, p = .0294; see Figure 1 – Figure supplement 1 for full behavioural statistics). THP decreased the effect of incentives on velocity (incentive * THP: β = -0.0216, p = .0030), indicating that muscarinic blockade diminished motivation by incentives. Figure 1d shows that this effect was similar in distractor absent/present trials, although slightly stronger when the distractor was absent; the 3-way (distractor*incentive*THP) interaction was not significant (p > .05), suggesting that the distractor-present trials had the same effect but weaker (Figure 1d).

Saccadic RT (time to initiation of saccade) was slower when participants were given THP (β = 0.0244, p = < .0001), faster with incentives (Figure 1e; β = -0.0767, p < .0001), and slowed by distractors (β = 0.0358, p < .0001). Again, THP reduced the effects of incentives (incentive*THP: β = 0.0218, p = .0002). Figure 1e shows that this effect was similar in distractor absent/present trials, although slightly stronger when the distractor was present; as the 3-way (distractor*incentive*THP) interaction was not significant and the direction of effects was the same in the two, it suggests the effect was similar in both conditions. Additionally, the THP*Incentive interactions were correlated between saccadic RT and residual velocity at the participant level (Figure 1 – Figure supplement 2).

We have given more details of the analyses performed in the Methods section and the results, as requested by you and the other reviewers (page 20, line 602):

Behavioural and EEG analysis included all 20 participants, although trials with EEG artefacts were included in the behavioural analyses (18585 trials in total) and not the EEG analyses (16627 trials in total), to increase power in the former. Removing these trials did not change the findings of the behavioural analyses.

We used single-trial linear-mixed effects models to analyse our data, including participant as a random effect of intercept, with the formula ‘~1 + incentive*distractor*THP + (1 | participant)’. We z-scored all factors to give standardised beta coefficients.

For the difference-wave cluster-based permutation tests (Figure 3 – Figure supplement 4), we used the DMGroppe Mass Univariate toolbox (Groppe et al., 2011), with 2500 permutations, to control the family-wise error rate at 0.05. This was used for looking at difference waves to test the effects of incentive, THP, and the incentive*THP interaction (using difference of difference-waves), across all EEG electrodes.

We adapted this toolbox to also run cluster-based permutation regressions to examine the relationship between the behavioural variables and the voltages at all EEG electrodes at each time point. On each iteration we shuffled the voltages across trials within each condition and person, and regressed it against the behavioural variable, with the model ‘~1 + voltage + incentive*distractorPresent*THP + (1 | participant)’. The Voltage term measured the association between voltage and the behavioural variable, after controlling for effects of incentive*distractor*THP on behaviour. By shuffling the voltages, we removed the relationship to the behavioural variable, to build the null distribution of t-statistics across electrodes and time-samples. We used the ‘cluster mass’ method (Bullmore et al., 1999; Groppe et al., 2011; Maris & Oostenveld, 2007) to build the null distribution, and calculated the p-value as the proportion of this distribution further from zero than the true t-statistics (two-tailed test). Given the relatively small sample size here, these whole-brain analyses should not be taken as definitive.

For the mediation analysis, we followed the 4-step process (Baron & Kenny, 1986; Muller et al., 2005), which requires 4 tests be met for the outcome (behavioural variable, e.g. RT), mediator (ERP, e.g., CNV) and the treatment (Incentive):

(1) Outcome is significantly associated with the Treatment (RT ~ 1 + Incentive + (1 | participant))

(2) Mediator is significantly associated with the Treatment (ERP ~ 1 + Incentive + (1 | participant))

(3) Mediator is significantly associated with the Outcome (RT ~ 1 + Incentive + ERP + (1 | participant))

(4) And the inclusion of the Mediator reduces the association between the Treatment and Outcome (Incentive effect from model #3)

The mediation was measured by the reduction in the absolute standardised beta coefficient between incentive and behaviour when the ERP mediator was included (model #3 vs model #1 above). We used permutation-testing to quantify the likelihood of finding these mediations under the null hypothesis, achieved by shuffling the ERP across trials (within each participant) to remove any link between the ERP and behaviour. We repeated this 2500 times to build a null distribution of the change in absolute beta-coefficients for the RT ~ Incentive effect when this permuted mediator was included (model #3 vs model #1). We calculated a one-tailed p-value by finding the proportion of the null distribution that was equal or more negative than the true value (as Mediation is a one-tailed prediction). For this mediation analysis, we only included trials with valid ERP measures, even for the models without the ERP included (e.g., model #1), to keep the trial-numbers and degrees of freedom the same.

Mediated moderation (Muller et al., 2005) was used to see whether the effect of THP (the Moderator) on behaviour is mediated by the ERP, with the following tests (after the previous Mediation tests were already satisfied):

(5) THP moderates the Incentive effect, via a significant Treatment*Moderator interaction on the Outcome (RT ~ 1 + Incentive + THP + Incentive*THP + (1 | participant))

(6) THP moderates the Incentive effect on the Mediator, via a Treatment*Moderator interaction on the Outcome (ERP ~ 1 + Incentive + THP + Incentive*THP + (1 | participant))

(7) THP’s moderation of the Incentive effect is mediated by the ERP, via a reduction in the association of Treatment*Moderator on the Outcome when the Treatment*Moderator interaction is included (RT ~ 1 + Incentive + THP + Incentive*THP + ERP + ERP*THP + (1 | participant))

Mediated moderation is measured as the reduction in absolute beta-coefficients for ‘RT ~ Incentive*THP’ between model #5 and #7, which captures how much of this interaction could be explained by including the Mediator*Moderator interaction (ERP*THP in model #7). We tested the significance of this with permutation testing as above, permuting the ERP across trials (within participants) 2500 times, and building a null distribution of the change in the absolute beta-coefficients for RT ~ Incentive*THP between models #7 and #5. We calculated a one-tailed p-value from the proportion of these that were equal or more negative than the true change.

(2) Please explain why only men were included in this study. We are all hoping that men-only research is a practice of the past.

We only included men to prevent any chance of administering the drug to someone pregnant. Trihexyphenidyl is categorized by the FDA as a Pregnancy Category Class C drug, and the ‘Summary of Product Characteristics’ states: “There is inadequate information regarding the use of trihexyphenidyl in pregnancy. Animal studies are insufficient with regard to effects on pregnancy, embryonal/foetal development, parturition and postnatal development. The potential risk for humans is unknown. Trihexyphenidyl should not be used during pregnancy unless clearly necessary.”

While the drug can be prescribed where benefits may outweigh this risk, as there were no benefits to participants in this study, we only recruited men to keep the risk at zero.

We have updated the Methods/Drugs section to explain this (page 17, line 494):

“The risks of Trihexyphenidyl in pregnancy are unknown, but the Summary Product of Characteristics states that it “should not be used during pregnancy unless clearly necessary”. As this was a basic research study with no immediate clinical applications, there was no justification for any risk of administering the drug during pregnancy, so we only recruited male participants to keep this risk at zero.”

And we have referenced this in the Methods/Participants section (page 18, line 501):

“Our sample size calculations suggested 27 participants would detect a 0.5 effect size with .05 sensitivity and .8 power. We recruited 27 male participants (see Drugs section above)”

(3) Please explain acronyms (eg EEG) when first used.

Thank you for pointing this out, we have explained EEG at first use in the abstract and the main text, along with FWER, M1r, and ERP which had also been missed at first use.

**Reviewer #3 (Recommendations For The Authors):**
The authors say: "Therefore, acetylcholine antagonism reduced the invigoration of saccades by incentives, and increased the pull of salient distractors. We next asked whether these effects were coupled with changes in preparatory neural activity." But I found this statement to be misleading since the primary effects of the drug seem to have been to decrease the frequency of distractor-repulsed saccades... so "decreased push" would probably be a better analogy than "increased pull".

Thank you for noticing this, we agree, and have changed this to (page 5, line 165):

“Therefore, acetylcholine antagonism reduced the invigoration of saccades by incentives, and decreased the repulsion of salient distractors. We next asked whether these effects were coupled with changes in preparatory neural activity.”

I don't see anything in EEG preprocessing about channel rejection and interpolation. Were these steps performed? There are very few results related to the full set of electrodes.

We did not reject or interpolate any channels, as visual inspection found no obvious outliers in terms of noisiness, and no channels had standard deviations (across time/trials) higher than our standard cutoff (of 80). The artefact rejection was applied across all EEG channels, so any trials with absolute voltages over 200uV in any channel were removed from the analysis. On average 104/120 trials were included (having passed this check, along with eye-movement artefact checks) per condition per person, and we have added the range of these, along with totals across conditions to the Analysis section and a statement about channel rejection/interpolation (page 20, line 588):

“Epochs were from -200:1500ms around the preparation cue onset, and were baselined to the 100ms before the preparation cue appeared. Visual inspection found no channels with outlying variance, so no channel rejection or interpolation was performed. We rejected trials from the EEG analyses where participants blinked or made saccades (according to EyeLink criteria above) during the epoch, or where EEG voltage in any channel was outside -200:200μV (muscle activity). On average 104/120 trials per condition per person were included (SD = 21, range = 21-120), and 831/960 trials in total per person (SD=160, range=313-954). A repeated-measures ANOVA found there were no significant differences in number of trials excluded for any condition (p > .2).”